# λ-Orthogonality Regularization for Compatible Representation Learning

**Simone Ricci**[1,2]*    **Niccolò Biondi**[1,2]    **Federico Pernici**[1,2]

**Ioannis Patras**[3]    **Alberto Del Bimbo**[1,2]

[1]DINFO (Department of Information Engineering), University of Florence, Italy
[2]MICC (Media Integration and Communication Center)
[3]Queen Mary University of London, UK

## Abstract

Retrieval systems rely on representations learned by increasingly powerful models. However, due to the high training cost and inconsistencies in learned representations, there is significant interest in facilitating communication between representations and ensuring compatibility across independently trained neural networks. In the literature, two primary approaches are commonly used to adapt different learned representations: affine transformations, which adapt well to specific distributions but can significantly alter the original representation, and orthogonal transformations, which preserve the original structure with strict geometric constraints but limit adaptability. A key challenge is adapting the latent spaces of updated models to align with those of previous models on downstream distributions while preserving the newly learned representation spaces. In this paper, we impose a relaxed orthogonality constraint, namely λ-Orthogonality regularization, while learning an affine transformation, to obtain distribution-specific adaptation while retaining the original learned representations. Extensive experiments across various architectures and datasets validate our approach, demonstrating that it preserves the model's zero-shot performance and ensures compatibility across model updates. Code available at: https://github.com/miccunifi/lambda_orthogonality.

## 1 Introduction

Retrieval tasks are increasingly relevant in real-world applications such as face recognition [1, 2, 3], image localization [4, 5, 6], and object identification [7, 8, 9]. In image retrieval, a gallery of labeled images is matched to query images to identify related ones, ideally of the same class. Instead of high-dimensional images, retrieval uses low-dimensional feature vectors obtained from embedding models. Enhancing retrieval performance often involves updating embedding models [10, 11] to leverage more expressive network architectures [12], new training techniques (e.g., loss functions) or training paradigms [13, 14, 15]. However, neural networks rarely produce compatible features, even when trained on the same data with identical methods and architectures [16]. Consequently, matching the features of new queries with those of older galleries can degrade retrieval performance due to incompatibility [15]. To address this, replacing the gallery features generated by the old model with those produced by the new model—a computationally expensive process known as backfilling—is required. The challenge of updating a base model while ensuring its backward compatibility and avoiding backfilling has been extensively investigated [17, 15, 18, 19, 20, 21]. Furthermore, the

---

*Corresponding author: `simone.ricci@unifi.it`

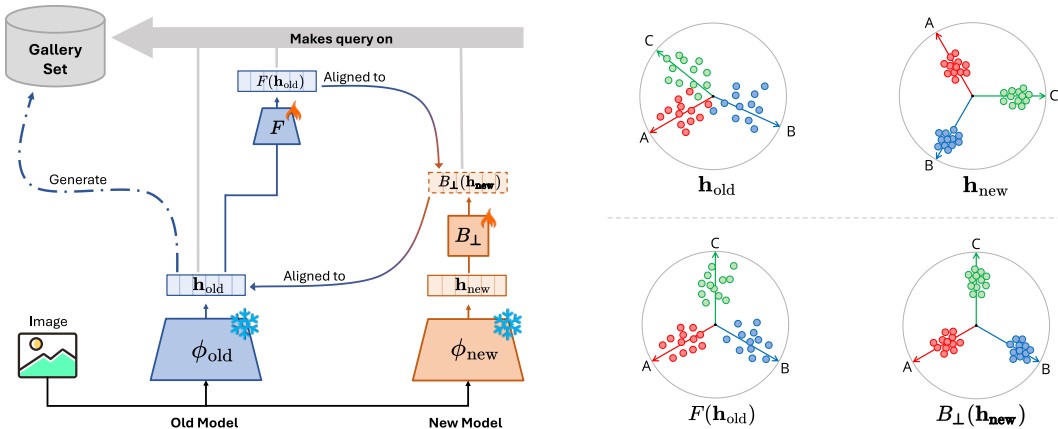

Figure 1: Overview of the proposed approach for achieving representation compatibility during retrieval system updates. A newly independently trained model is aligned to the old representation space via an orthogonal transformation $B_\perp$, which preserves geometric structure. A forward transformation $F$ maps the old representations to the backward-aligned space of the new model. Only the transformation parameters are optimized during training, while model parameters remain fixed.

optimal strategy for gallery updates—known as partial backfilling—has recently begun to receive attention [22].

Architectural changes and additional losses to ensure compatibility can reduce the performance of the updated model [23, 24]. To address this issue, research has focused on aligning the representation of a base model with that of an improved independently trained model using parameter-efficient adapters [22, 25]. On the other hand, the manifold hypothesis [26, 27] suggests that neural networks typically produce latent space representations of identical data distributions that differ primarily by a transformation. Consequently, mapping one representation to another requires only a few parameters, as functionally equivalent models approximate the same latent manifold [28, 29, 27]. Thus, a simple transformation aligning the new representation space to the previous one can provide the backward compatibility of the updated model.

Recent studies have focused on affine and orthogonal mappings to adapt the latent space of a base model (source space) to that of another model (target space), using specific data points as reference [30, 28, 31]. Within the plasticity-stability paradigm [32], affine mappings offer high adaptability (plasticity) but may alter the source space's configuration [33, 34]. Conversely, orthogonal mappings maintain the source space's geometric structure (stability), though they offer no adaptability to a different distribution. To preserve the geometric structure of the source space, particularly when it is more informative than the target space [28, 35], while enabling adaptability, we propose a novel regularization term. Different from previous work [36], our term constrains a transformation to remain within a specified proximity to the orthogonality condition, controlled by a hyperparameter $\lambda$.

In this paper, we address the challenge of ensuring compatibility between independently trained models by learning different transformations across representation spaces, as illustrated in Fig. 1. Our contributions are summarized as follows:

- We propose $\lambda$-Orthogonality regularization, a relaxed orthogonality constraint that retains the original representation space's global structure while enabling slight local adaptations for downstream tasks.

- We enhance representation compatibility by employing a supervised contrastive loss, which promotes intra-class clustering and inter-model alignment of feature representations, while remaining agnostic to model architecture.

- We conduct extensive experiments across diverse architectures and datasets, demonstrating that our method not only ensures compatibility between models but also promotes the preservation of the base model's latent space geometry, resulting in improved accuracy on downstream tasks.

- We propose a novel architecture-agnostic backfilling strategy that improves retrieval performance while optimizing the gallery update process.

## 2 Related Works

As demonstrated by [16], feature representations from two models—even if trained on the same data—do not generally coincide, creating costly backfilling in retrieval systems. To avoid this, [15] introduced Backward Compatible Training (BCT), which keeps the old classifier fixed as a reference so new embeddings align with prior class prototypes. Additionally, they provided a formal definition of compatibility between model representations. Subsequent research has expanded on this foundation, incorporating additional regularization techniques to better align new representations with previous ones [21, 37, 20, 38, 39] and implementing specific architecture design [18, 13, 40]. However, the performance of the updated backward-compatible models frequently falls to reach that of models trained independently [23], a consequence of the regularization imposed to achieve compatibility. To avoid this, [23] and [24] suggested expanding the representation space to include new classes while ensuring that the representations of old classes remain aligned during updates. To ensure compatibility between models trained independently, mapping-based strategies have been developed [41, 42, 43]. Forward Compatible Training (FCT), as detailed by [25], introduces a function that aligns embeddings from an older model to those of a newer model's space, incorporating additional side information for each data point. As noted by [25], the computational overhead of these transformations is minimal compared to the demands of processing images through the embedding model. FastFill [22] improves forward transformation learning by using a new model classifier and proposes a Bayesian strategy to optimize the gallery backfilling process leveraging the new model. In contrast, we propose a set of transformation functions to ensure not only forward but also backward compatibility during model updates, with a particular focus on the orthogonality property in backward mappings. Additionally, we propose a supervised contrastive loss that promotes intra-class clustering and inter-modality alignment, thereby enhancing adaptation. Finally, we propose a novel gallery backfilling strategy based on a distance metric that directly operates on pre-extracted gallery representations, making it agnostic to the underlying architecture.

## 3 Method

To achieve compatible representations between independently trained models, we introduce a theoretically grounded pipeline composed of multiple transformations. First, in Sec. 3.1 we report the definition of compatibility introduced by [15]. In Sec. 3.2 and 3.3, we introduce a novel backward-ompatibility method, which aligns the new model's representations to those of the previous model using either a strict orthogonal transformation or when adapting to a downstream task a transformation regularized by our proposed $\lambda$-Orthogonality constraint. Next, in Sec. 3.4, we present forward trasformation learning, which aligns the previous model's representations to those of the newly adapted model via an affine or more complex transformation, enabling effective gallery set updates. We also apply a supervised contrastive loss (Sec. 3.5) during transformation training to improve alignment between model representations and enhance intra-class cluster compactness, thereby satisfying the compatibility criterion defined in Def. 3.1. Finally, in Sec. 3.6, we propose a novel ordering strategy for backfilling the gallery with improved representations in an optimized sequence. Throughout our methodology, all models serve as fixed feature extractors with frozen parameters, while only the transformation layers are trained.

### 3.1 Backward-Compatible Representations Definition

The formulation of Backward-Compatibility between representations, introduced by [15], is closely related to the concept of latent space communication between different models [30]. The formal definition of backward-compatible representations specifies:

**Definition 3.1** (*Backward-Compatibility*). The representation of a model learned at step $k$ is compatible with the representation of a distinct model learned at a subsequent step $t$, where $k < t$, if the following condition is satisfied:

$$\forall i, j : \left(y_i = y_j \implies d(\mathbf{h}_i^t, \mathbf{h}_j^k) \leq d(\mathbf{h}_i^k, \mathbf{h}_j^k)\right) \wedge \left(y_i \neq y_j \implies d(\mathbf{h}_i^t, \mathbf{h}_j^k) \geq d(\mathbf{h}_i^k, \mathbf{h}_j^k)\right) \quad (1)$$

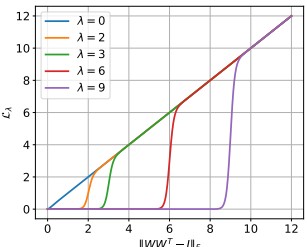
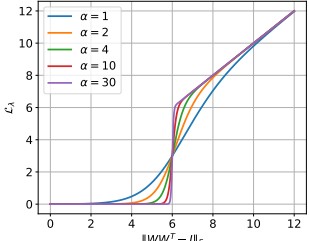
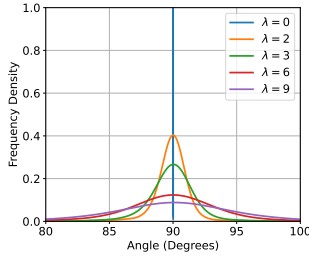

(a) Value of Eq. 6 at different $\lambda$.    (b) Effect of different $\alpha$ at $\lambda = 6$.    (c) KDE of $W$ angles.

Figure 2: Impact of $\lambda$-Orthogonality regularization on affine transformations. Fig. 2a shows the variation of Eq. 6 for different values of $\lambda$, demonstrating the influence of the threshold in the regularization. Fig. 2b illustrates the effect of varying $\alpha$ while keeping $\lambda = 6$, highlighting its behavior in the sigmoid function. Fig. 2c presents the kernel density estimation (KDE) of angles between the columns of matrix $W$ for different values of $\lambda$, showcasing the impact of regularization on orthogonality preservation.

where $d(\cdot, \cdot)$ is a distance function and $y_i$ and $y_j$ are the class labels associated with the extracted representation vectors $\mathbf{h}_i$ and $\mathbf{h}_j$, respectively. The inequalities in Def. 3.1 indicate that the new model's representation, when compared against the old representation, should perform at least as well as the previous model's in clustering images from the same class and separating them from those of different classes.

## 3.2 Backward Transformation

One of the contributions of relative encoding [30] is the observation that representation spaces, in practice, often differ only by an angle-preserving transformation when they share the same or similar data semantics. Furthermore, [28] demonstrates that when there is a difference in learned semantics, a transformation that preserves both angles and distances—learned with Procrustes analysis [44]—yields superior performance in cross-architecture and cross-modality classification tasks than only angle-preserving mappings. A transformation $T$ is defined as an isometry if it preserves angles and distances between any two points $a$ and $b$ in the space. Formally, a mapping $T : \mathbb{R}^n \to \mathbb{R}^n$ is an isometry if the following condition holds: $\|T(a) - T(b)\|_2 = \|a - b\|_2, \quad \forall a, b \in \mathbb{R}^n$, where $\| \cdot \|_2$ denotes the Euclidean norm, or equivalently, a general distance metric in other spaces. We leverage this property to achieve backward-compatible representations, aligning the updated model's space with the base model's using an orthogonal transformation. This maintains a unified representation space across updates, preserving the geometric properties and performance of the updated model due to the isometric nature of the transformation.

Given a base model $\phi^k$ and its updated version $\phi^t$, with $k < t$, and their corresponding representation vectors $\mathbf{h}^k \in \mathbb{R}^d$ and $\mathbf{h}^t \in \mathbb{R}^n$, we learn an orthogonal transformation $B_\perp : \mathbb{R}^n \to \mathbb{R}^n$ that maps the embedding space of the updated model into the space of the base model. To enforce strict orthogonality, a generic transformation $B$ is parameterized as the matrix exponential of a skew-symmetric matrix $P$, such that $B = e^P$, where the upper triangular entries of $P$ are learnable parameters [45]. To enforce alignment between the updated and base representation spaces, we optimize the transformation $B_\perp$ by minimizing the Mean Squared Error loss between $\mathbf{h}^k$ and the transformed $\mathbf{h}^t$:

$$\mathcal{L}_B = ||B_\perp(\mathbf{h}^t) - \mathbf{h}^k||_2^2 \tag{2}$$

As the transformation $B_\perp$ is a square matrix, if the dimensionalities of the two representation spaces differ, the higher-dimensional feature vector is truncated to match the dimensionality of the smaller representation space.

## 3.3 $\lambda$-Orthogonality Regularization

A strict orthogonal constraint (high stability) on a transformation $B$ might not be ideal when model distributions vary from those on which the adapter is trained—the case of private models providing only their extracted embeddings to the user. Imposing such a constraint can limit the integration of

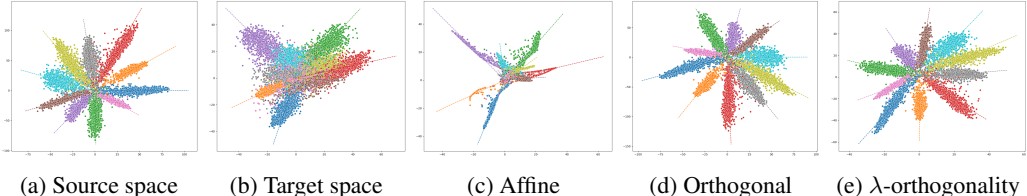

| (a) Source space | (b) Target space | (c) Affine | (d) Orthogonal | (e) $\lambda$-orthogonality |

Figure 3: Effects of affine (Fig. 3c), strictly orthogonal (Fig. 3d), and $\lambda$-orthogonality (with $\lambda = 1$) regularized (Fig. 3e) transformations trained to align a source representation space (Fig. 3a) learned with a LeNet model (embedding dimension = 2) on the complete MNIST dataset, with a target representation space (Fig. 3b) learned on the first five classes of MNIST using the same architecture.

new, relevant information for downstream tasks. Conversely, an affine transformation (high plasticity) without geometric regularization may disrupt the updated model's representations [46, 47]. As described in [36], a soft orthogonality constraint can be applied to the transformation $B : \mathbb{R}^n \to \mathbb{R}^n$, consisting of a weight matrix $W \in \mathbb{R}^{n \times n}$ and a bias term $b \in \mathbb{R}^n$. Previous works [48, 49, 50] have proposed constraining the Gram matrix of the weight matrix to be close to the identity matrix by minimizing a loss function defined as:

$$\mathcal{L}_{orth} = ||W^T W - I||_F \tag{3}$$

where $|| \cdot ||_F$ denotes the Frobenius norm and $W$ is the weight of the transformation $B$. This can be interpreted as a weight decay term that restricts the set of parameters to lie close to a Stiefel manifold [50]. However, this approach does not provide control over the specific level of orthogonality that can be imposed on the transformation.

To this end, we introduce a threshold $\lambda$, which specifies the desired proximity of the Gram matrix of the weight matrix to the identity matrix. A naive solution would be to stop the optimization of the $\mathcal{L}_{orth}$ loss once the Gram matrix of the weight matrix reaches the threshold $\lambda$, as the loss directly influences the Gram matrix:

$$\min_W ||W^T W - I||_F \quad \text{s.t.} \quad ||W^T W - I||_F \geq \lambda \tag{4}$$

This objective can be achieved directly through the use of a Heaviside step function [51, 52], shifted by the parameter $\lambda$: $H(x - \lambda) = \mathbf{1}_{\{x \geq \lambda\}}$ This function $H$ offers an efficient mechanism to control the degree of orthogonality during the minimization process, that effectively deactivating the regularization term in Eq. 3 when the Frobenius norm exceeds the threshold $\lambda$:

$$\mathcal{L}_\lambda = H(||WW^T - I||_F - \lambda) \cdot ||WW^T - I||_F. \tag{5}$$

However, this approach introduces a discontinuity in the loss function, as highlighted by [53]. In particular, their work focuses on evaluating the closeness of these sigmoid functions to the Heaviside step function, providing precise upper and lower bounds for the Hausdorff distance. Building on their theoretical and empirical analysis, we propose a smooth modulating function that ensures the effect of the constraint is gradually adjusted, with the penalty becoming more or less significant depending on the distance from the threshold $\lambda$. Specifically, we formulate a novel $\lambda$-Orthogonality Regularization term by optimizing a loss function defined as:

$$\mathcal{L}_\lambda = \sigma \left( \alpha \left( ||WW^T - I||_F - \lambda \right) \right) \cdot ||WW^T - I||_F \tag{6}$$

where $\sigma(\cdot)$ is the sigmoid function, and $\alpha$ is a scaling factor. The sigmoid function acts as a continuous switch that gradually turns the regularization term on and off near the value of $\lambda$, as shown in Fig. 2a. Instead, the scaling factor $\alpha$ controls the steepness of the sigmoid function, which in turn determines how sharply the regularization is activated or deactivated as the value of $||WW^T - I||_F$ approaches the threshold $\lambda$. In Fig. 2b, we illustrate different levels of steepness applied to the regularization loss. As $\alpha$ increases, its behavior converges more closely to the Heaviside step function.

To further analyze the behavior of the $\lambda$-orthogonality regularization, we optimize Eq. 6, applied to a transformation $B$ with a randomly initialized weight matrix $W$. As shown in Fig. 2c, the kernel density estimation (KDE) of these angles changes based on the value of $\lambda$ used in the regularization. Smaller values of $\lambda$ result in column vectors that become increasingly orthogonal, specifically when

$\lambda = 0$ our regularization is equal to Eq.3. Fig. 3 illustrates the effects of an affine (Fig. 3c), strictly orthogonal (Fig. 3d), and $\lambda$-orthogonality regularized (Fig. 3e) transformations trained to align a source representation space (Fig. 3a) learned on the full MNIST dataset with a target representation space (Fig. 3b) learned on the first five classes of MNIST. The toy experiment shows that the $\lambda$-orthogonal constraint improves alignment by relaxing strict orthogonality while encouraging preservation of the source feature space structure, in contrast to an unconstrained transformation.

## 3.4 Forward Transformation

In addition to a backward transformation that maps the representations of the new model to those of the previous model, it is possible to formulate a forward transformation $F : \mathbb{R}^d \rightarrow \mathbb{R}^n$. This transformation maps the representation vector $\mathbf{h}^k \in \mathbb{R}^d$ of the previous model to $\mathbf{h}^t \in \mathbb{R}^n$, the representation of the new model. Since the representation of the new model is superior to that of the previous model, the transformation $F$ should be affine (high plasticity) or multiple projection layers to better adapt to the improved representation. The transformation $F$ is learned by minimizing the Mean Squared Error between the two representations, as $||F(\mathbf{h}^k) - \mathbf{h}^t||_2^2$, following the approach described in [25]. This concept is closely related to latent space communication [30, 31], where $d(\mathbf{h}_i^k, \mathbf{h}_j^k) = d(\mathcal{T} \, \mathbf{h}_i^t, \mathcal{T} \, \mathbf{h}_j^t)$ with $\mathcal{T}$ is a generic transformation. In previous approaches [25, 22] the old representations $\mathbf{h}^k$ are aligned directly with the new $\mathbf{h}^t$ through transformation $F$, but incompatibility arise between $\mathbf{h}^k$ and $F(\mathbf{h}^k)$. As mentioned in Sec. 3.2, a backward orthogonal transformation $B_\perp$ realigns new representations with the old ones. Instead of adapting old features directly to new representations $\mathbf{h}^t$, we adapt them to $B_\perp(\mathbf{h}^t)$, ensuring a unified alignment across model updates. Furthermore, the transformations $F$ and $B_\perp$ can be trained jointly, as they utilize the same training data. Consequently, the forward alignment loss in our methodology is defined as:

$$\mathcal{L}_F = ||F(\mathbf{h}^k) - B_\perp(\mathbf{h}^t)||_2^2 \tag{7}$$

If the extracted representations are derived from a dataset different from the training sets of the two models, as discussed in Sec. 3.3, a $\lambda$-orthogonal regularized transformation $B_\lambda$ can be employed in place of the strictly orthogonal $B_\perp$.

## 3.5 Intra-class Clustering and Inter-Model Alignment

As discussed in Sec. 3.1, the compatibility inequalities defined in Def. 3.1 require not only alignment but also a higher concentration of clusters to achieve compatibility. To this end, [22] introduces an additional training loss, $\mathcal{L}_{disc}$, that, unlike the influence loss in [15], relies directly on the new model's classifier rather than the old one. However, $\mathcal{L}_{disc}$ depends on access to the new model's classifier and training loss, limiting its applicability, especially when the new model's architecture is unknown (e.g., embedding vectors from private or online models). To overcome this, we propose the use of a supervised contrastive loss, applied directly to representation vectors. This loss requires no classifier or architectural knowledge, as it directly leverages representation vectors for alignment and clustering. The supervised contrastive loss [54] minimizes the cross-entropy loss between $\mathbf{q}_i$ and $\mathbf{p}_i$:

$$\mathcal{L}_{\text{contr}} = -\sum_{i=1}^{K} \mathbf{p}_i \log \mathbf{q}_i \tag{8}$$

where $\mathbf{q}_i$ denotes the probability assigned to sample $i$ by applying a temperature-scaled softmax over the dot-product similarities between the L2-normalized feature $\mathbf{h}$ and each other candidate, and $\mathbf{p}_i$ is the normalized ground-truth indicator distribution that places equal mass on all semantically matching (same-class) candidates and zero on all others. Specifically, we utilized a combination of this loss function, where the objective $\mathcal{L}_C$ is defined as:

$$\mathcal{L}_C = \mathcal{L}_{\text{contr}}(F(\mathbf{h}^k), B_\perp(\mathbf{h}^t)) + \mathcal{L}_{\text{contr}}(F(\mathbf{h}^k), \mathbf{h}^k) \tag{9}$$

This loss encourages clustering among the adapted representations while also aligning them with those of the previous model, thereby promoting intra-class clustering and inter-model alignment of feature representations.

The overall loss function of our framework is defined as a weighted sum of four components: the forward alignment loss $\mathcal{L}_F$, the backward alignment loss $\mathcal{L}_B$, the contrastive loss $\mathcal{L}_C$, and the $\lambda$-Orthogonality regularization term $\mathcal{L}_\lambda$. Formally, the total loss is expressed as:

$$\mathcal{L} = w_1 \cdot \mathcal{L}_F + w_2 \cdot \mathcal{L}_B + w_3 \cdot \mathcal{L}_C + \mathcal{L}_\lambda \tag{10}$$

Table 1: Compatibility evaluation on ImageNet1K under two scenarios: (a) Extending classes setting, (b) Architecture update setting. For each case (highlighted with different colors), we report CMC-Top1 and mAP metrics.

(a) Extending classes setting. Two models trained independently: $\phi_{\text{old}}$ on first 500 classes, and $\phi_{\text{new}}$ on full ImageNet1K. Both use ResNet-34 with an embedding dimension of 128.

| Method | Query/Gallery | CMC-Top1 | mAP |
|---|---|---|---|
| Ind. Train. | $\phi_{\text{old}}/\phi_{\text{old}}$ | 43.56 | 25.18 |
| | $\phi_{\text{new}}/\phi_{\text{old}}$ | 0.10 | 0.15 |
| | $\phi_{\text{new}}/\phi_{\text{new}}$ | 61.61 | 35.69 |
| FCT [25] | $F(\phi_{\text{old}})/\phi_{\text{old}}$ | 0.10 | 0.15 |
| | $F(\phi_{\text{old}})/F(\phi_{\text{old}})$ | 50.13 | 30.93 |
| | $\phi_{\text{new}}/F(\phi_{\text{old}})$ | 57.21 | 33.00 |
| FastFill [22] | $F(\phi_{\text{old}})/\phi_{\text{old}}$ | 0.10 | 0.15 |
| | $F(\phi_{\text{old}})/F(\phi_{\text{old}})$ | 50.63 | 31.48 |
| | $\phi_{\text{new}}/F(\phi_{\text{old}})$ | 57.21 | 33.19 |
| Ours | $F(\phi_{\text{old}})/\phi_{\text{old}}$ | **44.59** | **26.70** |
| | $F(\phi_{\text{old}})/F(\phi_{\text{old}})$ | **51.46** | **33.75** |
| | $B_{\perp}(\phi_{\text{new}})/F(\phi_{\text{old}})$ | **57.41** | **34.53** |
| | $B_{\perp}(\phi_{\text{new}})/\phi_{\text{old}}$ | **43.94** | **25.75** |
| | $B_{\perp}(\phi_{\text{new}})/B_{\perp}(\phi_{\text{new}})$ | 61.61 | 35.69 |

(b) Independently Pretrained Models setting: Two models trained independently on the full ImageNet1K dataset. The first model, $\phi_{\text{old}}$, is a ResNet-18, whereas the second, $\phi_{\text{new}}$, is a ViT-L-16.

| Method | Query/Gallery | CMC-Top1 | mAP |
|---|---|---|---|
| Ind. Train. | $\phi_{\text{old}}/\phi_{\text{old}}$ | 55.62 | 26.91 |
| | $\phi_{\text{new}}/\phi_{\text{old}}$ | 0.04 | 0.17 |
| | $\phi_{\text{new}}/\phi_{\text{new}}$ | 76.62 | 56.84 |
| FCT [25] | $F(\phi_{\text{old}})/\phi_{\text{old}}$ | 0.04 | 0.17 |
| | $F(\phi_{\text{old}})/F(\phi_{\text{old}})$ | 59.39 | 42.65 |
| | $\phi_{\text{new}}/F(\phi_{\text{old}})$ | 72.54 | 49.85 |
| FastFill [22] | $F(\phi_{\text{old}})/\phi_{\text{old}}$ | 0.04 | 0.17 |
| | $F(\phi_{\text{old}})/F(\phi_{\text{old}})$ | **61.17** | **46.28** |
| | $\phi_{\text{new}}/F(\phi_{\text{old}})$ | 73.33 | **52.83** |
| Ours | $F(\phi_{\text{old}})/\phi_{\text{old}}$ | **60.83** | **40.69** |
| | $F(\phi_{\text{old}})/F(\phi_{\text{old}})$ | 61.10 | 45.91 |
| | $B_{\perp}(\phi_{\text{new}})/F(\phi_{\text{old}})$ | **73.53** | 52.06 |
| | $B_{\perp}(\phi_{\text{new}})/\phi_{\text{old}}$ | **65.54** | **38.55** |
| | $B_{\perp}(\phi_{\text{new}})/B_{\perp}(\phi_{\text{new}})$ | 76.62 | 56.84 |

where $w_1$, $w_2$, and $w_3$ denote scalar weights used to balance the contributions of each term.

## 3.6 Partial Backfilling Strategy

Determining an effective ordering for backfilling samples in the forward-adapted gallery set, where $F(\mathbf{h}^k)$ from the old model is replaced by $B_{\perp}(\mathbf{h}^t)$, is critical for achieving the performance of the new independently trained model as efficiently as possible. However, identifying the optimal ordering of backfilling represents a computationally intractable combinatorial problem [22]. To address this challenge, FastFill [22] introduces an ordering inspired by Bayesian Deep Learning. This approach models the alignment error as a multivariate Gaussian distribution and minimizes the negative log-likelihood of this distribution during the training of the mapping function $F$. However, from a retrieval perspective, the most representative instances—those that significantly enhance the separation between distinct classes—are identified as the embeddings closest to their respective class means [55, 56]. Accordingly, prioritizing the backfilling of the least informative embeddings will increase the system's performance by reinforcing class distinctions. To this end, we propose a novel method for estimating a backfill ordering based directly on the already extracted representation vector $F(\mathbf{h}^k)$. First, we calculate the mean representation vector $\boldsymbol{\mu}_c$ for each class $c$ in the forward-adapted gallery set. Then, we compute a distance metric $d$ of each embedding vector $F(\mathbf{h}^k)$ from its corresponding class mean $\boldsymbol{\mu}_c$. For instance, $d$ can be the Mean Squared Error, $d = \|F(\mathbf{h}^k) - \boldsymbol{\mu}_c\|_2$. Gallery embedding exhibiting the largest distance $d$ from $\boldsymbol{\mu}$ are prioritized for backfilling, thereby facilitating the matching with queries generated by the new backward-adapted independently trained model $B_{\perp}(\mathbf{h}^t)$.

## 4 Experiments

### 4.1 Image Retrieval Compatibility

Backward compatibility is crucial in retrieval tasks involving a gallery set $\mathcal{G} = \{(\mathbf{x}_i, y_i)\}_{i=1}^{N_g}$ and a query set $\mathcal{Q} = \{(\mathbf{x}_i, y_i)\}_{i=1}^{N_q}$, each containing $N_g$ and $N_q$ images respectively, with associated class labels. A base model indexes the gallery by extracting feature vectors from the images, which are then used to match with vectors from the query set in retrieval tasks. The compatibility definition presented in Def. 3.1 involves computing pairwise distances between all datapoints in the dataset. This process becomes increasingly computationally demanding as the dataset size grows. Then, a model updated at step $t$ is considered backward-compatible with the base model trained at step $k$ if

Table 2: Compatibility results for two models pretrained on ImageNet1K and adapted to downstream tasks: $\phi_{\text{old}}$, a ResNet-18, and $\phi_{\text{new}}$, a ViT-L-16, using as backward adapter $B_\lambda$ with $\lambda = 12$. The ZS column indicates the CMC-Top1 performance increase on ImageNet1K, with values in parentheses indicating the increment compared to the newly independently trained model. Each Query/Gallery case is highlighted with a different color to facilitate comparison of results.

| Method | Query/Gallery | Dataset | | | |
| | | **CUB** | | **CIFAR100** | |
| | | CMC-Top1 | ZS | CMC-Top1 | ZS |
|---|---|---|---|---|---|
| Ind. Train. | $\phi_{\text{old}}/\phi_{\text{old}}$ | 44.82 | | 51.13 | |
| | $\phi_{\text{new}}/\phi_{\text{old}}$ | 0.4 | | 0.8 | |
| | $\phi_{\text{new}}/\phi_{\text{new}}$ | 71.78 | | 74.08 | |
| FCT [25] | $F(\phi_{\text{old}})/\phi_{\text{old}}$ | 0.04 | | 0.8 | |
| | $F(\phi_{\text{old}})/F(\phi_{\text{old}})$ | 51.10 | | 57.35 | |
| | $\phi_{\text{new}}/F(\phi_{\text{old}})$ | 62.13 | | 69.80 | |
| FastFill [22] | $F(\phi_{\text{old}})/\phi_{\text{old}}$ | 0.4 | | 0.8 | |
| | $F(\phi_{\text{old}})/F(\phi_{\text{old}})$ | 54.50 | | 66.17 | |
| | $\phi_{\text{new}}/F(\phi_{\text{old}})$ | 61.49 | | 67.23 | |
| Ours | $F(\phi_{\text{old}})/\phi_{\text{old}}$ | **51.12** | | **67.29** | |
| | $F(\phi_{\text{old}})/F(\phi_{\text{old}})$ | **59.92** | | **67.72** | |
| | $B_\lambda(\phi_{\text{new}})/F(\phi_{\text{old}})$ | **70.72** | | **72.08** | |
| | $B_\lambda(\phi_{\text{new}})/\phi_{\text{old}}$ | **60.64** | | **71.85** | |
| | $B_\lambda(\phi_{\text{new}})/B_\lambda(\phi_{\text{new}})$ | **75.44 (+3.66)** | **+0.025** | **78.23 (+4.15)** | **+0.112** |

the Empirical Compatibility Criterion [15] is satisfied:

$$M\left(\Phi_t^{\mathcal{Q}}, \Phi_k^{\mathcal{G}}\right) > M\left(\Phi_k^{\mathcal{Q}}, \Phi_k^{\mathcal{G}}\right), \quad \text{with } k < t \tag{11}$$

where $M$ denote a performance metric, $\Phi^{\mathcal{G}}$ and $\Phi^{\mathcal{Q}}$ represent the extracted gallery and query sets, respectively. Specifically, $M\left(\Phi_t^{\mathcal{Q}}, \Phi_k^{\mathcal{G}}\right)$ assesses cross-model retrieval with gallery features from the updated model at step $t$ and query features from step $k$. In contrast, $M\left(\Phi_k^{\mathcal{Q}}, \Phi_k^{\mathcal{G}}\right)$ refers to same-model retrieval, where both gallery and query features originate from the same model at step $k$.

**Partial Backfilling.** Given an ordering $\pi$ of the images in the gallery set $\Phi^{\mathcal{G}}$, denoted as $\mathbf{x}_{\pi_1}, \mathbf{x}_{\pi_2}, \ldots, \mathbf{x}_{\pi_n}$, and a backfilling fraction $\beta \in [0, 1]$, we define the partially backfilled gallery set $\Phi_{\pi,\beta}^{\mathcal{G}}$ as follows. The first $N_{g,\beta} = \lfloor \beta N_g \rfloor$ images in the ordering are processed using the updated model, while the remaining images are processed using the old model. Here, $N_g$ denotes the total number of images in the gallery. To evaluate different backfilling strategies, we employ the backfilling metric $\widetilde{M}$, introduced in [22], which is defined as: $\widetilde{M}(\Phi^{\mathcal{G}}, \Phi^{\mathcal{Q}}, \pi) = \mathbb{E}_{\beta \sim [0,1]} M(\Phi_{\pi,\beta}^{\mathcal{G}}, \Phi^{\mathcal{Q}})$. This metric is the area under the backfilling curve when evaluating performance using $M$.

## 4.2 Evaluation Metrics and Datasets

Following prior work on model compatibility [15, 25], we evaluate performance using two metrics. The Cumulative Matching Characteristics (CMC), which measures top-$k$ retrieval accuracy by computing distances between query and gallery features, considers retrieval successful if at least one of the $k$ closest gallery images shares the query's label. The mean Average Precision (mAP) measures the area under the precision-recall curve across the full recall range $[0, 1]$.

To validate our approach, we utilize the following datasets: ImageNet1K [57], CIFAR100 [58], and CUB200 [59]. Each dataset's validation/test set serves as both the query and gallery, with each query image removed from the gallery to avoid trivial matches during search. The notation 'Query/Gallery' indicates the models used for extracting embeddings in all tables, respectively. CUB200 and CIFAR100 are employed as downstream tasks.

## 4.3 Extending Classes Setting

In this setting, we update a base model by extending the number of classes. We train two models independently: $\phi_{\text{old}}$ on the first 500 classes and $\phi_{\text{new}}$ on all 1000 classes of ImageNet1K, both using a ResNet-34 architecture with an embedding dimension of 128, following PyTorch's standard training

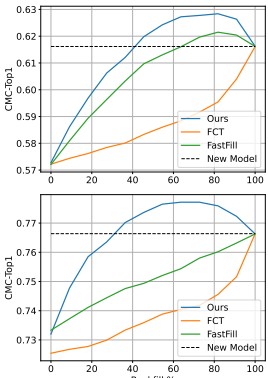
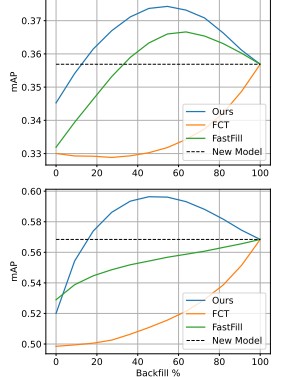

Table 3: Extended Classes setting

| Method | $\widetilde{M}$ | |
| --- | --- | --- |
| | CMC-Top1 | mAP |
| FCT [25] | 58.72 | 33.57 |
| FastFill [22] | 60.49 | 35.59 |
| Ours | **61.20** | **36.46** |

Table 4: Independently Pretrained Models setting

| Method | $M$ | |
| --- | --- | --- |
| | CMC-Top1 | mAP |
| FCT [25] | 73.86 | 52.02 |
| FastFill [22] | 75.06 | 55.34 |
| Ours | **76.59** | **57.72** |

Figure 4: Partial backfilling results for the Extending Classes setting (top Figures) of Tab. 1a, and Independently Pretrained Models setting (bottom Figures) of Tab. 1b. We use features from the new model $\phi_{\text{new}}$ for the query set (otherwise $B_\perp(\phi_{\text{new}})$ if trained). For the gallery set, we begin with forward-adapted old features $F(\phi_{\text{old}})$ and incrementally replace them with new features.

recipe[2]. After training the two models independently, adapters are optimized using Adam with a learning rate of $0.001$, while keeping the model layers frozen. We compare our method against FCT [25] and FastFill [22], two mapping methods used to achieve compatible representations. In Tab. 1a, the performance of each method is summarized following the metrics of Sec. 4.2. The results indicate that the new model, $\phi_{\text{new}}$, is not directly compatible with the old one $\phi_{\text{old}}$. Additionally, the two mapping methods, FCT and FastFill, enhance performance across both metrics for the adapted representations of the gallery and query sets. However, these methods achieve backward compatibility with the newly trained model but not with the original one. In contrast, our method aligns the new model with the old one through the orthogonal transformation $B_\perp$. This ensures compatibility between the new and old representations and also enhances the performance provided by the forward adapter $F$. Appendix A provides additional results on Places365 [60] dataset.

## 4.4 Independently Pretrained Models adapted on Downstream Task

Due to escalating training costs, pretrained models are increasingly used, especially for adapting to downstream tasks with local datasets. In this context, we employ two models—available in the PyTorch hub—pretrained on the ImageNet1K dataset: a ResNet-18 with an embedding size of 512, and a more advanced Vision Transformer (ViT-L-16) [61] with an embedding size of 1024. The ViT model is considered an update over the ResNet-18 due to its enhanced architecture. Tab. 1b shows adapter training results using the same dataset as the two pretrained models, revealing a trend similar to Tab. 1a and demonstrating our method's comparable performance to other baselines, but with compatibility between the updated model and the previous one. Unlike FastFill, our approach does not require the new model's classifier, relying directly on the extracted embedding vectors. In Appendix B, to further validate our method, we apply our approach to different architectures used as pretrained models. Instead, in Appendix C, we investigate update scenarios involving distribution or objective shifts using CLIP-like [62] models and self-supervised architectures such as DINOv2 [63].

Results for compatibility on downstream tasks are reported in Tab. 2, where adapters are trained on representations from local datasets (CUB200 or CIFAR100) different from the training dataset. Employing a transformation $B_\lambda$ with $\lambda$-Orthogonality regularization, our method enhances local task performance and model compatibility, outperforming the baselines. Results on additional downstream datasets (Flower102 [64] and Places365) are reported in Appendix D. From Tab. 1a and Tab. 1b, we observe that a strict orthogonal transformation, $B_\perp$, does not result in performance improvements relative to the independently trained model $\phi_{\text{new}}$. Conversely, $B_\lambda$, which provides more plasticity with respect to $B_\perp$, enables the new model to enhance performance in the downstream task. An ablation study on the hyperparameter $\lambda$ is presented in Appendix E, and a component-wise ablation of the loss terms in Eq. 3.5 is detailed in Appendix F.

---

[2]pytorch/vision/tree/main/references/classification

### 4.5 Backfilling Results

In this section, we evaluate our novel backfill strategy discussed in Sec. 3.6, considering the experimental setting detailed in Tab. 1a and Tab. 1b. Given that FCT lacks a specific backfilling strategy, we employ a random ordering as in [22]. The results, depicted in Fig. 4, Tab. 3, and Tab. 4, demonstrate that our backfilling strategy outperforms the other baselines by a certain margin. Notably, Fig. 4 illustrates that with less than 50% of the gallery backfilled, we achieve the same performance as the newly independently trained model. In Appendix G, we provide an ablation study using an alternative distance metric to the Mean Squared Error employed in the main experiments.

## 5    Conclusion

Model compatibility is a critical challenge in many large-scale retrieval systems and can hinder system updates when not achieved. In this paper, we introduce mapping transformations that align independently learned representations under a unified space, also providing a more feature clustering through supervised contrastive loss. We also propose a relaxation of the orthogonality constraint to aid adaptation to downstream tasks without compromising the integrity of newly trained independent models. Additionally, we propose a novel backfill ordering strategy that enables efficient partial backfilling of the gallery set, achieving the performance of a newly independently trained model with less than half of the gallery backfilled. Our approach demonstrates superior performance compared to previous methods, across the same and different data distributions on which the models are trained. To contextualize these results, the limitations of the approach are examined in detail in Appendix I. Furthermore, to evaluate its practical utility, we analyze its methodological complexity and broader applicability in Appendix H.

## Acknowledgments

This paper was partially funded by the project "Collaborative Explainable neuro-symbolic AI for Decision Support Assistant", CAI4DSA, CUP B13C23005640006.

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

Table 5: Compatibility evaluation on Places365 under the Extending Classes setting. We use two independently trained ResNet-50 models: $\phi_{\text{old}}$ trained on the first 205 classes, and $\phi_{\text{new}}$ trained on all classes of Places365.

| Method | Query/Gallery | CMC-Top1 | mAP |
|---|---|---|---|
| Ind. Train. | $\phi_{\text{old}}/\phi_{\text{old}}$ | 33.86 | 15.76 |
| | $\phi_{\text{new}}/\phi_{\text{old}}$ | 0.21 | 0.33 |
| | $\phi_{\text{new}}/\phi_{\text{new}}$ | 37.37 | 19.11 |
| FCT [25] | $F(\phi_{\text{old}})/\phi_{\text{old}}$ | 0.21 | 0.33 |
| | $F(\phi_{\text{old}})/F(\phi_{\text{old}})$ | 36.43 | 19.02 |
| | $\phi_{\text{new}}/F(\phi_{\text{old}})$ | 37.04 | 18.99 |
| FastFill [22] | $F(\phi_{\text{old}})/\phi_{\text{old}}$ | 0.21 | 0.33 |
| | $F(\phi_{\text{old}})/F(\phi_{\text{old}})$ | 39.71 | 23.98 |
| | $\phi_{\text{new}}/F(\phi_{\text{old}})$ | 38.42 | 19.94 |
| Ours | $F(\phi_{\text{old}})/\phi_{\text{old}}$ | **38.65** | **21.88** |
| | $F(\phi_{\text{old}})/F(\phi_{\text{old}})$ | **39.96** | **26.19** |
| | $B_\perp(\phi_{\text{new}})/F(\phi_{\text{old}})$ | **38.50** | **21.77** |
| | $B_\perp(\phi_{\text{new}})/\phi_{\text{old}}$ | 35.47 | 17.98 |
| | $B_\perp(\phi_{\text{new}})/B_\perp(\phi_{\text{new}})$ | 37.37 | 19.11 |

# A    Extending Classes Setting on Places365

To validate our approach further, we evaluate it using a model trained on a dataset different from ImageNet1K. Specifically, we use a ResNet-50 pretrained on Places205 (from ViSSL) as the old model, and a ResNet-50 pretrained on Places365 (from CSAILVision) as the new model. Tab. 5 summarizes the performance of each method using the evaluation metrics defined in Sec.4.2. The results demonstrate that the new model $\phi_{\text{new}}$ is not inherently compatible with the old model, $\phi_{\text{old}}$. Moreover, the adaptation $F(\phi_{\text{old}})$ provided by FCT underperforms when compared to the new model alone. In contrast, methods that promote better clustering, such as FastFill and our proposed approach, achieve even higher performance than the standalone new model. This improvement arises from leveraging information from both the old and new models, effectively implementing a form of knowledge distillation during the learning of the forward adapter. Unlike the baselines, our method aligns all adapted representations within a unified representation space, thereby consistently maintaining compatibility with the old model.

# B    Additional Architecture for Independently Pretrained Models Setting

We conduct additional experiments using a DenseNet-121 as the old model, $\phi_{\text{old}}$, and an EfficientNet-B3 as the new model, $\phi_{\text{new}}$, both pretrained on ImageNet1K and obtained from the PyTorch Hub. The results of these experiments on the ImageNet1K dataset are presented in Tab. 6. Our approach achieves the best performance across all metrics, outperforming the baselines in both cross-model and same-model retrieval scenarios.

# C    Additional Experiments with DINOv2 and CLIP as Independently Pretrained Models

To investigate update scenarios involving data distribution or objective shifts, we conduct additional experiments using a ResNet-18 pretrained on ImageNet1K as the old model, and both a CLIP [62] pretrained on CC12M [65] dataset and a DINOv2 [63] ($vit\_small\_patch14\_dinov2$) as the new models. To train both the forward and backward transformations, the ImageNet1K dataset and the same hyperparameters of Tab. 1b are used. This setup represents a considerable shift in both data distribution and model objective relative to the new models. Notably, FastFill cannot be applied in this context, as both CLIP and DINOv2 lack classifiers. In Tab. 7a, we report the results obtained using DINOv2 as the new, independently trained model. Our approach achieves better results than FCT, further validating its practical applicability to real-world problems.

Table 6: Compatibility results on ImageNet1K under the Independently Pretrained Models setting. We use two independently trained models: DenseNet-121 as the old model, $\phi_{\text{old}}$, and an EfficientNet-B3 as the new model, $\phi_{\text{new}}$, both pretrained on ImageNet1K and obtained from the PyTorch Hub.

| Method | Query/Gallery | CMC-Top1 | mAP |
|---|---|---|---|
| Ind. Train. | $\phi_{\text{old}}/\phi_{\text{old}}$ | 62.02 | 32.95 |
| | $\phi_{\text{new}}/\phi_{\text{old}}$ | 0.11 | 0.16 |
| | $\phi_{\text{new}}/\phi_{\text{new}}$ | 71.60 | 54.90 |
| FCT [25] | $F(\phi_{\text{old}})/\phi_{\text{old}}$ | 0.11 | 0.16 |
| | $F(\phi_{\text{old}})/F(\phi_{\text{old}})$ | 68.16 | 53.22 |
| | $\phi_{\text{new}}/F(\phi_{\text{old}})$ | 70.64 | 54.63 |
| FastFill [22] | $F(\phi_{\text{old}})/\phi_{\text{old}}$ | 0.11 | 0.16 |
| | $F(\phi_{\text{old}})/F(\phi_{\text{old}})$ | 67.76 | 57.22 |
| | $\phi_{\text{new}}/F(\phi_{\text{old}})$ | 69.47 | 57.43 |
| Ours | $F(\phi_{\text{old}})/\phi_{\text{old}}$ | **69.25** | **50.20** |
| | $F(\phi_{\text{old}})/F(\phi_{\text{old}})$ | **69.29** | **57.36** |
| | $B_\perp(\phi_{\text{new}})/F(\phi_{\text{old}})$ | **71.33** | **57.50** |
| | $B_\perp(\phi_{\text{new}})/\phi_{\text{old}}$ | **67.23** | **44.34** |
| | $B_\perp(\phi_{\text{new}})/B_\perp(\phi_{\text{new}})$ | 71.60 | 54.90 |

Table 7: Compatibility evaluation involving data distribution or objective shifts: (a) ResNet-18 as old model and DINOv2 [63] ($vit\_small\_patch14\_dinov2$) as the new models; (b) ResNet-18 as old model and CLIP [62] pretrained on CC12M [65] as the new model. For each case, we report CMC-Top1 and mAP metrics.

(a) DINOv2 [63]. A shift in the objective function is present between the old and new models.

| Method | Query/Gallery | CMC-Top1 | mAP |
|---|---|---|---|
| Ind. Train. | $\phi_{\text{old}}/\phi_{\text{old}}$ | 55.62 | 26.91 |
| | $\phi_{\text{new}}/\phi_{\text{old}}$ | 0.04 | 0.17 |
| | $\phi_{\text{new}}/\phi_{\text{new}}$ | 71.92 | 44.04 |
| FCT [25] | $F(\phi_{\text{old}})/\phi_{\text{old}}$ | 0.04 | 0.17 |
| | $F(\phi_{\text{old}})/F(\phi_{\text{old}})$ | 59.33 | 37.53 |
| | $\phi_{\text{new}}/F(\phi_{\text{old}})$ | 67.97 | 41.07 |
| Ours | $F(\phi_{\text{old}})/\phi_{\text{old}}$ | **54.82** | **32.14** |
| | $F(\phi_{\text{old}})/F(\phi_{\text{old}})$ | **61.30** | **41.95** |
| | $B_\perp(\phi_{\text{new}})/F(\phi_{\text{old}})$ | **68.74** | **43.78** |
| | $B_\perp(\phi_{\text{new}})/\phi_{\text{old}}$ | **58.73** | **31.50** |
| | $B_\perp(\phi_{\text{new}})/B_\perp(\phi_{\text{new}})$ | 71.92 | 44.07 |

(b) CLIP [62]. A shift in the objective function and the data distribution is present between the old and new models.

| Method | Query/Gallery | CMC-Top1 | mAP |
|---|---|---|---|
| Ind. Train. | $\phi_{\text{old}}/\phi_{\text{old}}$ | 55.62 | 26.91 |
| | $\phi_{\text{new}}/\phi_{\text{old}}$ | 0.04 | 0.17 |
| | $\phi_{\text{new}}/\phi_{\text{new}}$ | 44.29 | 16.15 |
| FCT [25] | $F(\phi_{\text{old}})/\phi_{\text{old}}$ | 0.04 | 0.17 |
| | $F(\phi_{\text{old}})/F(\phi_{\text{old}})$ | 42.58 | 16.93 |
| | $\phi_{\text{new}}/F(\phi_{\text{old}})$ | 42.96 | 16.88 |
| Ours | $F(\phi_{\text{old}})/\phi_{\text{old}}$ | **61.13** | **41.22** |
| | $F(\phi_{\text{old}})/F(\phi_{\text{old}})$ | **57.69** | **41.08** |
| | $B_\perp(\phi_{\text{new}})/F(\phi_{\text{old}})$ | **44.93** | **29.26** |
| | $B_\perp(\phi_{\text{new}})/\phi_{\text{old}}$ | **30.02** | **16.68** |
| | $B_\perp(\phi_{\text{new}})/B_\perp(\phi_{\text{new}})$ | 44.29 | 16.15 |

Instead, in Tab. 7b, we report the results obtained using CLIP pretrained on CC12M as the new, independently trained model. In this scenario, the pretrained CLIP model exhibits lower retrieval performance on ImageNet1K compared to ResNet-18. This is a well-known limitation of multi-modal training, where intra-modal misalignment can negatively impact the quality of single-modality representations [66]. Specifically, CLIP models are optimized for cross-modal retrieval rather than single-modality retrieval tasks, in contrast to DINOv2 or ResNet-18, which are trained exclusively on a single modality. This reduction in performance of the new model relative to the old one causes FCT to degrade the overall retrieval capacity of the system, failing to achieve compatibility, as it attempts to transform the higher-quality representations of the old model into the lower-performing representations of the new model. In contrast, our method introduces an additional loss that encourages both intra-class clustering and inter-model alignment of feature representations on the specific training dataset. As a result, the forward transformation, due to its greater flexibility, improves the performance of the old model's representations. Even in this challenging scenario, our approach outperforms FCT, further validating the robustness of our method.

Table 8: Compatibility results on Places365 and Flowers102 for two models pretrained on ImageNet1K and adapted to downstream tasks: $\phi_{\text{old}}$, a ResNet-18, and $\phi_{\text{new}}$, a ViT-L-16, using a backward adapter $B_\lambda$ with $\lambda = 12$. The ZS column indicates the CMC-Top1 performance increase on ImageNet1K, with values in parentheses showing the increment compared to the newly independently trained model.

| Method | Query/Gallery | Places365 | | Flowers102 | |
|---|---|---|---|---|---|
| | | CMC-Top1 | ZS | CMC-Top1 | ZS |
| Ind. Train. | $\phi_{\text{old}}/\phi_{\text{old}}$ | 22.41 | | 84.35 | |
| | $\phi_{\text{new}}/\phi_{\text{old}}$ | 0.20 | | 1.20 | |
| | $\phi_{\text{new}}/\phi_{\text{new}}$ | 35.15 | | 99.39 | |
| FCT [25] | $F(\phi_{\text{old}})/\phi_{\text{old}}$ | 0.20 | | 1.20 | |
| | $F(\phi_{\text{old}})/F(\phi_{\text{old}})$ | 28.17 | | 86.71 | |
| | $\phi_{\text{new}}/F(\phi_{\text{old}})$ | 32.12 | | 99.07 | |
| FastFill [22] | $F(\phi_{\text{old}})/\phi_{\text{old}}$ | 0.20 | | 1.20 | |
| | $F(\phi_{\text{old}})/F(\phi_{\text{old}})$ | 26.38 | | 53.78 | |
| | $\phi_{\text{new}}/F(\phi_{\text{old}})$ | 33.04 | | 11.12 | |
| Ours | $F(\phi_{\text{old}})/\phi_{\text{old}}$ | **28.84** | | **83.36** | |
| | $F(\phi_{\text{old}})/F(\phi_{\text{old}})$ | **29.80** | | **89.90** | |
| | $B_\lambda(\phi_{\text{new}})/F(\phi_{\text{old}})$ | **33.27** | | **99.41** | |
| | $B_\lambda(\phi_{\text{new}})/\phi_{\text{old}}$ | **29.94** | | **98.17** | |
| | $B_\lambda(\phi_{\text{new}})/B_\lambda(\phi_{\text{new}})$ | **36.38 (+1.23)** | **+0.38** | **99.54 (+0.15)** | **+0.01** |

## D  Additional Datasets for Independently Pretrained Models adapted on Downstream Task setting

We further extend our analysis of the Independently Pretrained Models Adapted on Downstream Task setting by including two additional datasets: the larger Places365 and the fine-grained Flowers102. These additions allow us to evaluate our method's effectiveness in more challenging scenarios. The results are reported in Tab. 8. In these experiments, the old model is a ResNet-18 and the new model is a ViT-L-16, both pretrained on ImageNet-1K. We employ an affine adapter with $\lambda = 12$. On both additional datasets, our approach consistently outperforms the baseline methods. The proposed $\lambda$-Orthogonality regularization not only improves retrieval performance on the downstream tasks but also encourages the adapted new model representation, $B_\lambda(\phi_{\text{new}})$, to remain consistent with its original form. As a result, retrieval performance on ImageNet1K is preserved.

## E  Ablation on the hyperparameter $\lambda$

In our experiments, we select $\lambda$ to maximize adaptability to downstream tasks while preserving the pre-trained model's performance on its original training dataset, ImageNet1K. To illustrate the impact of our approach, Tab. 9 reports the CMC-Top1 scores obtained by applying our proposed $\lambda$-orthogonal regularizer to the new pre-trained model. The results, also reported in Fig. 5, indicate that increasing $\lambda$ enhances the performance of the new model's representations on the downstream task.

However, this improvement comes at the expense of reduced performance on the original dataset, as evidenced by a decrease in zero-shot (ZS) scores, particularly pronounced in the absence of regularization ($\lambda = \infty$). Empirically, we find that setting $\lambda = 12$ yields the best trade-off across all metrics. [36] optimize a soft orthogonality constraint, equal to case where $\lambda = 0$. However, this formulation does not lead to performance improvements and is outperformed by

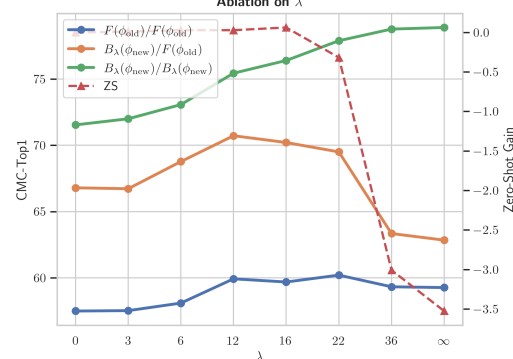

Figure 5: Ablation on our $\lambda$-orthogonal regularization on CUB dataset. Displayed are the compatibility metrics on CUB and the zero-shot (ZS) improvement on ImageNet1K at different value of $\lambda$. Results correspond to those in Tab. 9.

Table 9: Ablation over orthogonal regularization strength $\lambda$ on CUB dataset. Compatibility metrics on the target task and zero-shot (ZS) CMC-Top1 gain on ImageNet1K. Parentheses show the increment in CMC-Top1 over the independently trained new model on CUB dataset.

| $\lambda$ | $F(\phi_{\text{old}})/F(\phi_{\text{old}})$ | $B_\lambda(\phi_{\text{new}})/F(\phi_{\text{old}})$ | $B_\lambda(\phi_{\text{new}})/B_\lambda(\phi_{\text{new}})$ | ZS |
|---|---|---|---|---|
| $\perp$ (strict orth.) | 57.52 | 66.89 | 71.78 (+0.000) | +0.000 |
| 0 | 57.49 | 66.79 | 71.54 (−0.241) | −0.001 |
| 3 | 57.52 | 66.72 | 72.00 (+0.224) | +0.008 |
| 6 | 58.09 | 68.77 | 73.07 (+1.294) | +0.028 |
| **12** | 59.92 | **70.72** | 75.44 (+3.659) | +0.028 |
| 16 | 59.68 | 70.21 | 76.40 (+4.625) | **+0.062** |
| 22 | **60.20** | 69.50 | 77.89 (+6.109) | −0.318 |
| 36 | 59.32 | 63.34 | 78.77 (+6.990) | −3.008 |
| $\infty$ (no reg.) | 59.26 | 62.84 | **78.89** (+7.110) | −3.526 |

Table 10: Comparison of orthogonal regularization methods with different weight scales $w$. Compatibility metrics on the downstream task CUB200 and zero-shot (ZS) CMC-Top1 gain on ImageNet1K. Parentheses show the increment in CMC-Top1 over the independently trained new model. The last column reports the final value of $\|W^\top W - I\|_F$.

| $w$ | Method | $F(\phi_{\text{old}})/F(\phi_{\text{old}})$ | $B_\lambda(\phi_{\text{new}})/F(\phi_{\text{old}})$ | $B_\lambda(\phi_{\text{new}})/B_\lambda(\phi_{\text{new}})$ | ZS | $\|W^\top W - I\|_F$ |
|---|---|---|---|---|---|---|
| 1 | SO | 57.48 | 66.79 | 71.54 (−0.241) | −0.001 | 0.09 |
| 1 | SRIP | 57.38 | 66.57 | 71.66 (−0.120) | −0.001 | 0.08 |
| 1 | **Ours** ($\lambda = 12$) | **59.92** | **70.72** | **75.44 (+3.659)** | **+0.028** | **12.05** |
| $10^{-1}$ | SO | 59.11 | 69.56 | 74.88 (+3.106) | +0.022 | 9.50 |
| $10^{-1}$ | SRIP | 58.88 | 63.58 | 78.77 (+6.990) | −1.467 | 29.55 |
| $10^{-1}$ | **Ours** ($\lambda = 12$) | **59.93** | **70.70** | **75.20 (+3.419)** | **+0.076** | **12.12** |
| $10^{-2}$ | SO | 59.06 | 63.54 | 79.06 (+7.283) | −1.344 | 29.27 |
| $10^{-2}$ | SRIP | 59.23 | 63.42 | 78.73 (+6.955) | −3.077 | 35.42 |
| $10^{-2}$ | **Ours** ($\lambda = 12$) | 59.06 | 63.54 | 79.06 (+7.283) | −1.344 | 29.27 |
| $10^{-3}$ | SO | 58.71 | 62.91 | 78.78 (+7.007) | −3.162 | 35.54 |
| $10^{-3}$ | SRIP | 58.83 | 63.18 | 78.92 (+7.145) | −3.457 | 38.63 |
| $10^{-3}$ | **Ours** ($\lambda = 12$) | 58.71 | 62.91 | 78.78 (+7.007) | −3.162 | 35.54 |

the use of a strictly orthogonal transformation. As discussed in Sec. 3.3, imposing strict orthogonality may hinder the model's ability to incorporate task-specific information. In contrast, our approach relaxes this constraint by introducing a tunable hyperparameter $\lambda$ that controls the deviation of the Gram matrix from the identity, allowing greater flexibility while preserving representational consistency.

To further validate our aproach we also study the effect of a scalar weight $w$ to the loss contributions of our $\lambda$-orthogonal regularization compared with two different orthogonal regularizations: Soft Orthogonality (SO)[36]—witch correspond to the spacial case of $\lambda = 0$ in our aproach— and Spectral Restricted Isometry Property (SRIP)[36]. We test the regularizers across different values of scalar weight: $w = 1$, $w = 10^{-1}$, $w = 10^{-2}$, and $w = 10^{-3}$. Additionally, we include a column reporting the exact value of $\|W^\top W - I\|_F$ reached by the backward transformation $B_\lambda$ at the end of training, to indicate the deviation from strict orthogonality.

As shown in the Tab. 10, for both SRIP and SO, the final value of $\|W^\top W - I\|_F$ is governed by the optimization process and the chosen scalar weight $w$. Unlike our $\lambda$-orthogonal regularization, these approaches do not provide direct control over $\|W^\top W - I\|_F$; a smaller contribution of the regularizer to the total loss results in a diminished regularization effect on the backward transformation $B_\lambda$. When the scalar weight $w$ of the regularizer is reduced, the optimization process is unable to fully minimize the regularization term, particularly because competing loss components (such as MSE and the contrastive loss $L_C$) may favor a non-orthogonal transformation. For instance, when $w = 10^{-3}$ and $w = 10^{-2}$, the results obtained with SO, SRIP, and our $\lambda$-orthogonal regularization are comparable to those observed in the case of $\lambda = \infty$ (see Tab. 9), where the orthogonality constraint is entirely ignored. This occurs because, at such a small value of $w$, the contribution of the regularizer becomes negligible during optimization. To avoid this issue, in our method we set $w = 1$ for the $\lambda$-orthogonal regularization, thereby ensuring that the regularization term is effectively

incorporated into the optimization process during the training of the backward transformation. This ensures that the regularization term achieves the target threshold $\lambda$, enabling precise control over the stability–plasticity trade-off in the backward transformation and leads to higher representation compatibility on the downstream task. As highlighted by the bold entries in the Tab. 10, our method produces stable results (minor fluctuations are attributable to stochastic optimization) for $w = 1$ and $w = 10^{-1}$ in contrast to SO and SRIP. Conversely, when $w$ is very low ($10^{-2}$ or $10^{-3}$), the regularizer cannot be fully optimized, and our method behaves similarly to SO regularization, as our introduced constrains ($\|W^\top W - I\|_F \geq \lambda$) influences the minimum of the objective, which is never reached in practice. In contrast, due to its approximate formulation and greater complexity relative to SO, SRIP exhibits an even weaker regularization effect when $w$ is low.

## F    Detailed Analysis of Loss Term Contributions

In this section, we analyze the contribution of each term to the final loss (Eq. 3.5) optimized during training. Tab. 11 presents the results obtained when the adaptation dataset matches the dataset used to train the models from which the features were extracted, namely ImageNet1K. In this scenario, a strict orthogonal transformation $B_\perp$ is employed for backward-compatibility. We observe that when used independently, $\mathcal{L}_F$ ensures compatibility with the representations of the new model but significantly fails to achieve backward compatibility. This behavior highlights a pronounced forward bias inherent to $\mathcal{L}_F$. The backward alignment loss $\mathcal{L}_B$ alone promotes backward compatibility but degrades forward-adapted representation performance. The contrastive loss $\mathcal{L}_C$ alone significantly improves inter-model alignment and intra-class clustering, supporting both backward and forward compatibility. The combination $\mathcal{L}_F + \mathcal{L}_B + \mathcal{L}_C$ achieves the highest overall performance across compatibility scenarios, underscoring the importance of each loss component in maintaining balance between forward and backward trasformation learning.

Tab. 12 shows the impact of these loss terms in a downstream task setting (CUB dataset), where $\phi_{old}$ is ResNet-18 and $\phi_{new}$ is ViT-L-16, using $\lambda$-Orthogonality with $\lambda = 12$. Similar to Tab. 11, excluding the backward loss $\mathcal{L}_B$ still yields good forward compatibility but significantly reduces backward compatibility performance. Excluding the contrastive loss $\mathcal{L}_C$ substantially decreases the adaptation to the downstream task leading to lower $B_\lambda(\phi_{\text{new}})/B_\lambda(\phi_{\text{new}})$ values. Using all loss terms $\mathcal{L}_F + \mathcal{L}_B + \mathcal{L}_C$ consistently achieves the best or near-best results in forward and backward compatibility, demonstrating the complementary nature of these terms.

These analyses underline that each loss term contributes uniquely and significantly to achieving comprehensive and model compatibility across various tasks.

Table 11: CMC-Top1 (%) on ImageNet1K for different loss combinations ($\checkmark$ = included, $\times$ = excluded). The setting is the same of Tab. 1b, where the first model, $\phi_{\text{old}}$, is a ResNet-18, whereas the second, $\phi_{\text{new}}$, is a ViT-L-16.

| Losses | | | Query/Gallery (CMC-Top1 %) | | | | |
|---|---|---|---|---|---|---|---|
| $\mathcal{L}_F$ | $\mathcal{L}_B$ | $\mathcal{L}_C$ | $F(\phi_{\text{old}})/\phi_{\text{old}}$ | $F(\phi_{\text{old}})/F(\phi_{\text{old}})$ | $B_\perp(\phi_{\text{new}})/\phi_{\text{old}}$ | $B_\perp(\phi_{\text{new}})/F(\phi_{\text{old}})$ | $B_\perp(\phi_{\text{new}})/B_\perp(\phi_{\text{new}})$ |
| $\checkmark$ | $\times$ | $\times$ | 0.04 | 59.09 | 0.04 | 72.27 | 76.63 |
| $\times$ | $\checkmark$ | $\times$ | 0.04 | 49.34 | 62.75 | 0.04 | 76.63 |
| $\times$ | $\times$ | $\checkmark$ | 61.24 | 58.63 | 64.97 | 60.83 | 76.63 |
| $\checkmark$ | $\checkmark$ | $\times$ | 54.18 | 59.29 | 62.77 | 72.46 | 76.63 |
| $\checkmark$ | $\times$ | $\checkmark$ | **61.25** | 60.43 | 65.13 | 73.44 | 76.63 |
| $\times$ | $\checkmark$ | $\checkmark$ | 60.85 | 59.09 | 65.42 | 57.90 | 76.63 |
| $\checkmark$ | $\checkmark$ | $\checkmark$ | 60.83 | **61.10** | **65.54** | **73.53** | 76.63 |

## G    Distance metric for Partial Backfilling Ordering

Our proposed partial backfilling strategy is guided by a distance metric $d$, which measures the dissimilarity between each embedding vector $F(\mathbf{h}^k)$ and its corresponding class mean $\boldsymbol{\mu}_c$. This section investigates the impact of different distance metrics on determining an effective ordering for backfilling images in the gallery set. We compare two distance metrics—Mean Squared Error (MSE) and Cosine Distance—for ranking images during partial backfilling. The performance of each metric is evaluated under two distinct experimental conditions: the Extending Classes setting (Tab. 13) and

Table 12: CMC-Top1 (%) on CUB for different loss combinations (✓ = included, × = excluded). The setting is the same of Tab. 2, where the first model, $\phi_{\text{old}}$, is a ResNet-18, whereas the second, $\phi_{\text{new}}$, is a ViT-L-16. A backward adapter $B_\lambda$ with $\lambda = 12$ is used to adapt the improved model on the downstream task.

| Losses | | | Query/Gallery (CMC-Top1 %) | | | | |
|---|---|---|---|---|---|---|---|
| $\mathcal{L}_F$ | $\mathcal{L}_B$ | $\mathcal{L}_C$ | $F(\phi_{\text{old}})/\phi_{\text{old}}$ | $F(\phi_{\text{old}})/F(\phi_{\text{old}})$ | $B_\lambda(\phi_{\text{new}})/\phi_{\text{old}}$ | $B_\lambda(\phi_{\text{new}})/F(\phi_{\text{old}})$ | $B_\lambda(\phi_{\text{new}})/B_\lambda(\phi_{\text{new}})$ |
| ✓ | × | × | 0.0 | 51.72 | 0.0 | 63.82 | 72.14 |
| × | ✓ | × | 0.0 | 35.27 | 45.80 | 0.0 | 71.91 |
| ✓ | ✓ | × | 37.15 | 47.56 | 46.56 | 60.70 | 69.76 |
| × | × | ✓ | 52.79 | 59.14 | 58.38 | 66.46 | 73.36 |
| ✓ | × | ✓ | 50.43 | 59.88 | 58.57 | 70.13 | 74.86 |
| × | ✓ | ✓ | **53.27** | 58.66 | 60.45 | 59.44 | 73.12 |
| ✓ | ✓ | ✓ | 51.12 | **59.92** | **60.64** | **70.72** | **75.44** |

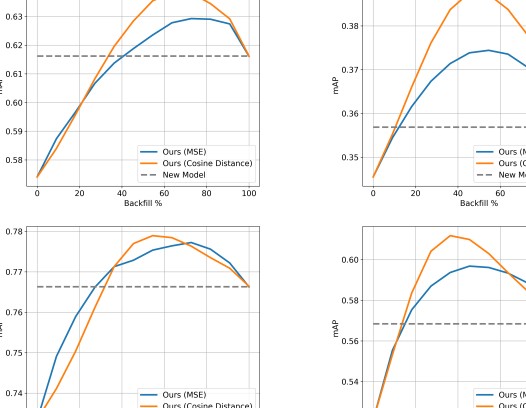

Table 13: Extended Classes setting

| Method | $\widetilde{M}$ | |
|---|---|---|
| | CMC-Top1 | mAP |
| MSE | 61.20 | 36.46 |
| Cosine Distance | **61.68** | **37.10** |

Table 14: Independently Pretrained Models setting

| Method | $\widetilde{M}$ | |
|---|---|---|
| | CMC-Top1 | mAP |
| MSE | **76.59** | 57.72 |
| Cosine Distance | 76.49 | **58.18** |

Figure 6: Different distance metric ablation for our partial backfilling strategy. Results for the Extending Classes setting (top Figures) of Tab. 1a, and Independently Pretrained Models setting (bottom Figures) of Tab. 1b. We use features from the new backward-adapted model $B_\perp(\phi_{\text{new}})$ for the query set. For the gallery set, we begin with forward-adapted old features $F(\phi_{\text{old}})$ and incrementally replace them with new features.

the Independently Pretrained Models setting (Tab. 14). MSE computes the Euclidean distance between feature vectors, capturing both angular and magnitude discrepancies. As shown in Tab. 13 and Tab. 14, MSE generally yields robust performance, particularly in terms of CMC-Top1. In contrast, Cosine Distance measures the angular distance between normalized feature vectors, emphasizing directional similarity while ignoring magnitude. The results indicate that Cosine Distance achieves slightly better performance in terms of mAP and provides comparable CMC-Top1 scores relative to MSE.

# H   Method Complexity and Broader Applicability

**Method Complexity.**   Our approach requires training only two matrices, resulting in a small number of parameters to optimize. Because our method operates solely on the extracted embeddings, it does not require any knowledge of the underlying models and is therefore applicable across different objectives (see Appendix C), architectures, and types of learned representations.

In contrast to previous methods, which either focus solely on alignment loss without any representation clustering loss (e.g., FCT [25]), or require specific architectural components of the pretrained models (e.g., FastFill [22], which requires access to the classifier of the new model), our approach addresses these limitations. Additionally, while existing baselines provide only forward adaptation, our method is designed to achieve both forward and backward compatibility, thereby addressing practical needs that prior works do not meet. For instance:

- $B_\perp(\phi_\text{new})/F(\phi_\text{old})$ yields higher retrieval values compared to the baselines.
- $B_\perp(\phi_\text{new})/\phi_\text{old}$ can be achieved exclusively by our method. From a practical standpoint, this allows compatibility to be established even before all gallery items are forward-adapted using $F$.
- Since our approach provides a unified representation space, even when the gallery is in a hybrid form (i.e., with some elements already adapted by $F$ and others not), using $B_\perp(\phi_\text{new})$ still ensures compatibility. This can not be achieved neither by FCT [25] nor FastFill [22].

The contrastive loss defined in Eq. 8 relies on the availability of class labels to encourage embeddings from the same class to cluster together while pushing apart embeddings from different classes. In scenarios where class labels are not available, Eq. 8 naturally reduces to an unsupervised contrastive loss, similar to the objective used for training CLIP models [62]. In this unsupervised setting, we contrast pairs of representations originating from different models, and clustering—since it cannot be enforced directly—becomes a byproduct resulting from embedding similarity. Consequently, our approach is flexible and can be applied in both supervised and unsupervised training scenarios, depending on the availability of labels for the downstream task.

**Broader Applicability.** As it is demonstrated in [36], soft orthogonalization has been applied to regularize all the weights of a CNN during training, and could benefit from the increased plasticity offered by our proposed $\lambda$-orthogonal regularization. While retrieval is the standard scenario for evaluating compatibility [15], our approach is broadly applicable to any task that requires representation adaptation, as it focuses on model alignment and clustering of learned representations. As demonstrated in our downstream task adaptation experiments (see Sec. 4.4), our regularization approach yields improved performance compared to a strict orthogonal constraint, making it a valuable approach in domain adaptation scenarios as well. Furthermore, enforcing geometrical consistency while allowing adaptability has recently been investigated in the context of continual learning for multimodal training [67]. However, the authors of [67] promote this property indirectly through a knowledge consolidation loss, rather than by directly applying a regularization constraint. This highlights both possible future research and the potential applicability of our $\lambda$-orthogonal regularization across various areas of representation learning.

# I  Limitations

Our approach relies on the assumption that the new model's embedding space is more expressive (e.g., higher retrieval accuracy, stronger clustering) than that of the old model. If the updated model is not comparable or lower quality, due, for instance, to domain mismatch, insufficient training data, or architectural regressions, then both the forward and backward adapters may fail to improve performance or could even degrade compatibility. In many practical systems, this assumption is justified by scaling laws [68, 69, 70, 71] (i.e., larger models and more data generally yield better feature representations). For downstream tasks adaptation, while our $\lambda$-orthogonal regularized adapter shows strong performance and compatibility across various retrieval tasks, a manual tuning of the orthogonality threshold ($\lambda$) is needed. The trade-off between preserving the original model's geometry and allowing sufficient plasticity to adapt to new data hinges critically on the choice of $\lambda$. In practice, this hyperparameter could be selected via cross-validation or a small hyperparameter search on a held-out portion of the downstream dataset. Although we found that $\lambda = 12$ provides a good balance in our experiments (Appendix E), different downstream domains (e.g., fine-grained vs. coarse categories) and adapted representations may require different tuning of $\lambda$ to achieve optimal performance. Automating or self-tuning this parameter remains an open challenge.

