# OpenReview forum: "$\boldsymbol{\lambda}$-Orthogonality Regularization for Compatible Representation Learning"
_NeurIPS.cc/2025/Conference — NeurIPS 2025 poster_

### Official Review · Reviewer_uQX7 · 2025-06-02

**Clarity:** 3
**Significance:** 3
**Originality:** 2
**Rating:** 4
**Confidence:** 4

**Summary:**

This paper addresses the important problem of adapting and aligning representations between independently trained retrieval models, a practical issue for maintaining model compatibility in real-world applications. The authors propose a method that introduces a relaxed orthogonality regularization during the learning of an affine transformation, aiming to balance adaptation flexibility with preservation of the original representation space. They provide extensive experiments across multiple architectures and datasets, demonstrating that the proposed method facilitates cross-model compatibility and maintains zero-shot performance.

**Questions:**

Could the authors clarify why they chose their specific form of relaxed orthogonality regularization? Have alternative forms like SRIP been considered or tested? Or simply multiply the original regularization term by a coefficient. Would simply lowering the standard orthogonality penalty achieve a similar effect to the proposed approach?

How does the proposed method compare, both theoretically and empirically, with existing methods such as (deep) canonical correlation analysis, which also aim to align representations while preserving information or distribution? The loss form of CCA is essentially the same as the MSE loss form proposed in this paper.

The loss functions for backward-compatibility and forward-adaptation seem very similar in terms of implementation. What practical differences arise from separating these objectives, and could one unified framework suffice?

How sensitive is the method to the regularization coefficient in the orthogonality constraint?

**Ethical Concerns:**

["NO or VERY MINOR ethics concerns only"]

**Final Justification:**

My concerns have been resolved.

**Limitations:**

The novelty of the proposed orthogonality regularization is limited, given the existence of many similar techniques in the literature that are not considered or compared.

The empirical evaluation, while extensive, does not include direct comparison with established alignment and compatibility methods such as (deep) CCA or recent hybrid approaches incorporating autoencoders and regularization.

The theoretical distinction between backward-compatibility and forward-adaptation losses is not matched by a clear practical difference, reducing the impact of this contribution.

**Quality:**

2

**Strengths And Weaknesses:**

Strengths:

The problem of backward and forward compatibility between model representations is highly relevant, particularly for retrieval systems that must remain usable across model updates.

The paper provides comprehensive empirical results on a variety of architectures and datasets, strengthening the validity of the findings.

The proposed method is conceptually simple and can be readily integrated into existing model adaptation pipelines.

Weaknesses:

The choice of orthogonality regularization and its specific relaxation is not sufficiently justified. There exist several alternative and arguably more principled forms of soft orthogonality constraints (e.g., spectral norm-based regularizations such as SRIP), which are neither discussed nor compared.

The distinction between their backward-compatibility and forward-adaptation objectives appears to be mainly conceptual; in practice, the loss implementations are almost identical, differing only in the direction of the transformation. This calls into question the novelty and practical benefit of splitting these objectives.

The paper overlooks substantial related work in representation alignment, especially canonical correlation analysis (CCA), deep CCA, and their regularized or autoencoder-augmented variants, which also aim to preserve information and distributional properties during alignment. The omission of discussion and comparison with these methods limits the positioning of the proposed approach in the broader literature.

---

> ### Author Rebuttal · Authors · 2025-07-31
>
> We thank the Reviewer for their insightful comments, which helped us highlight our contributions. We appreciate their recognition of our method’s relevance to representation compatibility in model updates, its conceptual simplicity, and ease of integration. Below we address the questions raised by the Reviewer.
>
> **[Novelty and Justification of the λ-Orthogonality Regularization]**
>
> The novelty of the λ-orthogonality regularization is that it allows for controlled local flexibility when adapting a pre-trained model to a downstream task. This distinguishes our approach from methods such as the Spectral Restricted Isometry Property (SRIP) and Soft Orthogonality (SO), which do not provide explicit control over the orthogonality constraint.
>
> SRIP [36] is a regularization method that uses the spectral norm to enforce soft orthogonality in a matrix $W$. The SRIP loss is defined as $L_{SRIP}=\sigma(W^T W - I)$, where $\sigma(\cdot)$ denotes the largest singular value. This penalizes the worst-case deviation, i.e., $\max_i|\sigma_i(W)^2-1|$. However, $\sigma$ is a non-smooth function, making the optimization of $L_{SRIP}$ more complex than optimizing a smooth objective. To reduce computational cost, [36] approximates the spectral norm using a two-step power-iteration method, which only provides an approximate solution.
>
> On the other hand, SO is a much faster alternative, defined as:
>
> $L_{orth}=||W^{\top}W-I||_F$
>
> i.e., the sum of deviations of all inner products from the identity. Its gradient has a closed form:
>
> $\nabla_W L_{orth}=\frac{2W(W^{\top}W-I)}{||W^{\top}W-I||_F}$.
>
> Since SO is simple and its gradient is explicit, we use it as a soft orthogonality constraint instead of SRIP. The Frobenius norm aggregates deviations across all inner products, so small changes in individual dimensions only slightly increase $L_{orth}$. This allows the transformation to distribute error across its entries, preserving overall proximity to the Stiefel manifold while still adapting the representation for downstream tasks.
>
> Motivated by this, we introduce a novel, controlled relaxation of $L_{orth}$ by constraining it with a threshold λ defined as:
> $$
> \min_{W}||W^TW-I||_F\quad\text{s.t.}\quad||W^TW-I||_F\geq\lambda
> $$
> This formulation allows for increased local flexibility modulated by λ, in contrast to other regularizations which do not provide granular control over the proximity to the orthogonality condition.
>
> We also compared our λ-orthogonal regularization with SO and SRIP using the experimental setup in Tab. 9 where the represetations of two pretrained models are adapted to a downstream task.
>
> |Method|$F(\phi_{\text{old}})/F(\phi_{\text{old}})$| $B_{\lambda}(\phi_{\text{new}})/F(\phi_{\text{old}})$| $B_{\lambda}(\phi_{\text{new}})/B_{\lambda}(\phi_{\text{new}})$ |ZS|
> |-|-|-|-|-|
> |SO (λ=0)|57.48|66.79|71.54 (–0.241)|–0.001|
> |SRIP|57.38|66.57|71.66 (–0.120)|–0.001|
> |**Ours (λ=12)**|**59.92**|**70.72**|**75.44 (+3.659)**|**+0.028**|
>
> We observe that SRIP achieves retrieval performance comparable to SO. As detailed in Appendix D, SO corresponds to the case of our proposed λ-orthogonality regularization when λ=0. This further justify the choice of SO over SRIP, as it is faster and does not require any approximatation [36].
>
> [36] Bansal et al., “Can we gain more from orthogonality regularizations in training deep networks?”, NeurIPS 2018.
>
> **[Does Scaling the Standard Orthogonality Penalty Is Equal to the λ‑Orthogonality?]**
>
> The Reviewer raises a very interesting point. However, the answer is no: simply multiplying the original regularization term by a coefficient that lowers the orthogonality penalty does not achieve the same effect as our λ-orthogonality regularization.
>
> The key difference lies in the regularization itself. The λ-orthogonality is formulated as the constrained optimization problem:
>
> $$
> \min_{W}||W^\top W-I||_F\quad\text{s.t.}\quad||W^{\top}W - I||_F\geq\lambda
> $$
>
> This is different from multiplying the soft orthogonality penalty by λ:
>
> $$
> \min_{W}\lambda||W^{\top}W-I||_F
> $$
>
> In the former case, the minimum of the constrained problem is λ, while in the latter, the minimum will always be 0, even when scaled by λ. Therefore, there is no direct control on the final value of $||W^\top W - I||_F$ and thus to the trade-off between plasticity and stability of the adaptation of the representation, that is instead demanded to the optimization process.
> Fig. 2 empirically shows this effect with a toy problem. It can be observed how λ influences both the angle between the columns of $W$ and the final value of $||W^{\top}W-I||_F$.
>
> **[Sensitivity of the proposed method on the λ value]**
>
> This is examined in Appendix D, where we perform an ablation study on the coefficient λ. Tab. 9 shows results for various λ values, while Fig. 5 demonstrates the trade-off between stability (strict orthogonality) and plasticity (no regularization); increasing λ improves downstream task performance but reduces zero-shot scores, especially when λ=$\infty$.
>
> **[Relation to CCA and Related Alignment Methods]**
>
> We thank the Reviewer for relating our method to statistical approaches like Canonical Correlation Analysis (CCA) and deep CCA. However, our approach is primarily inspired by and closely related to Procrustes analysis, which is empirically distinct from CCA as shown in [3]. To clarify our position, we summarize the main differences between Procrustes analysis and CCA.
>
>
> Procrustes analysis has played an important role in aligning latent spaces of DNNs [4,5,28]. It provides correspondences between latent spaces of different models by estimating an optimal orthogonal transformation $R$ that best matches two paired point sets $X$ and $Y$ [9]. This is done by minimizing the least-squares error $|XR-Y|^2$. In contrast, CCA finds pairs of weight vectors $(a_i,b_i)$ such that the linear projections $U_i=Xa_i$ and $V_i=Yb_i$ are maximally correlated and mutually uncorrelated, producing a series of canonical variable pairs with descending correlations ($\rho_1\geq\rho_2,\ldots$). Thus, while Procrustes yields a single transformation, CCA returns multiple pairs of canonical variables.
>
> Moreover, their respective objective functions differ: Procrustes minimizes a least-squares distance, whereas CCA maximizes correlation. As a result, the two methods generally yield different optima as CCA is invariant to arbitrary full-rank linear transformations applied individually to $X$ or $Y$, whereas Procrustes is only invariant to joint orthogonal transformations of both sets. CCA and Procrustes coincide only in specific cases of 1-d and when both $X$ and $Y$ are whitened and have the same dimension. However, whitening usually harms performance by removing variance structure that encodes semantic information in representations of pretrained models. Therefore, using CCA to estimate transformation matrices leads to suboptimal retrieval performance.
>
> Directly estimating the transformation $B$ via Procrustes analysis is, in principle, possible; however, it also limits the applicability of our method. Procrustes analysis yields only orthogonal mappings and is therefore suitable for determining the backward transformation. However, as discussed in Sec. 3.4, the forward transformation requires a degree of plasticity rather than stability (strict orthogonality). Furthermore, while Procrustes analysis is computed in a closed form, our approach necessitates batch-wise optimization as it involves other losses, such as the $L_C$. In the downstream adaptation setting, Procrustes analysis does not permit the slight modifications in the representation that our backward transformation, learned with λ-orthogonality regularization, allows. For these reasons, we adopt the loss functions discussed in Sec. 3.2, 3.3, and 3.4.
>
> [3] Ding et al., “Grounding representation similarity through statistical testing,” NeurIPS 2021.
>
> [4] Wang & Mahadevan, “Manifold alignment using procrustes analysis,” ICML 2008.
>
> [5] Wang & Mahadevan, “Manifold alignment without correspondence,” IJCAI 2009.
>
> [9] Gower, “Generalized procrustes analysis,” Psychometrika, 1975.
>
> [28] Maiorca et al., “Latent space translation via semantic alignment,” NeurIPS 2023.
>
> **[Separation of Backward and Forward Transformations Objectives]**
>
> The Reviewer is correct that both backward and forward transformations are learned with the same mean squared error (MSE) loss. However, while learned through the same function, the two objectives yield distinct optimization signals.
>
> In backward adaptation, only the adapter $B_{\perp}$ is optimized, aligning the new representation $\mathbf{h}^t$ to the old one ($\mathbf{h}^k$). Since $B_{\perp}$ is an isometric transformation, the geometry of $\mathbf{h}^t$ is preserved.
>
> In contrast, forward adaptation updates the old representation $\mathbf{h}^k$ via adapter $F$ to match $B_{\perp}(\mathbf{h}^t)$. Unlike $B_{\perp}$, $F$ is not constrained to be orthogonal. As a result, the forward transformation enables greater flexibility to adjust the old representation, potentially improving it, as also shown by FCT and FastFill.
>
> Appendix E details the effects of each loss component individually and all their combinations. Jointly optimizing $L_F$, $L_B$, and $L_C$ yields the best performance across compatibility scenarios. As shown in Tab. 10 and 11, using only $L_B$ or $L_F$ affects performance depending on the transformation: optimizing only $L_B$ results in poor performance for $F$, and vice versa. The key novelty of our work lies in learning these two transformations jointly, as prior work focused only on forward adaptation and did not address backward compatibility.

---

> > ### Comment · Reviewer_uQX7 · 2025-08-05
> >
> > Thank you for the detailed response during the rebuttal period. Could the authors also evaluate SO or SRIP as regularizers by fine-tuning their corresponding coefficients? The current results are presented with a weight of 1, which is quite a large value. I would expect a softer regularization effect with weights such as 1e-1, 1e-2, or 1e-3, and it would be insightful to see their performance under these conditions.

---

> ### Author Response · Authors · 2025-08-06
>
> We thank the reviewer for the thoughtful suggestion. We have included the requested experiments in the following table. The experimental setting is identical to that used in our previous response (as well as in Tables 9 and 2 of the manuscript). However, in this case, we apply a scalar weight $w$ to the loss contributions of SO, SRIPS, and our λ-orthogonal regularization. We compare the values $w = 1$, $w = 10^{-1}$, $w = 10^{-2}$, and $w = 10^{-3}$. Additionally, we include a column reporting the exact value of $\lVert W^{\top} W-I \rVert_F$ of the transformation $B_λ$ at the end of training, to indicate its deviation from strict orthogonality.
>
> |$w$|Method|$F(\phi_{\text{old}})/F(\phi_{\text{old}})$| $B_λ(\phi_{\text{new}})/F(\phi_{\text{old}})$| $B_λ(\phi_{\text{new}})/B_{λ}(\phi_{\text{new}})$ |ZS| Final Value of $\lVert W^{\top}W -I\rVert_F$|
> |-|-|-|-|-|-|-|
> |$1$|SO|57.48|66.79|71.54 (–0.241)|–0.001| 0.09 |
> |$1$|SRIP|57.38|66.57|71.66 (–0.120)|–0.001| 0.08 |
> |$1$|**Ours (λ=12)** |	**59.92** |**70.72** |**75.44 (+3.659)** |**+0.028**| **12.05**|
> |||||||
> |$10^{-1}$|SO|59.11|69.56|74.88 (+3.106)|+0.022| 9.50 |
> |$10^{-1}$|SRIP|58.88|63.58|78.77 (+6.990)|–1.467| 29.55|
> |$10^{-1}$|**Ours (λ=12)**|**59.93**|**70.70**|**75.20 (+3.419)**|**+0.076**| **12.12** |
> |||||||
> |$10^{-2}$|SO|59.06|63.54|79.06 (+7.283)|-1.344| 29.27 |
> |$10^{-2}$|SRIP|59.23|63.42|78.73 (+6.955)|-3.077| 35.42 |
> |$10^{-2}$|**Ours (λ=12)**|59.06|63.54|79.06 (+7.283)|-1.344| 29.27 |
> |||||||
> |$10^{-3}$|SO|58.71|62.91|78.78 (+7.007)|-3.162| 35.54 |
> |$10^{-3}$|SRIP|58.83|63.18|78.92 (+7.145)|-3.457| 38.63 |
> |$10^{-3}$|**Ours (λ=12)**|58.71|62.91|78.78 (+7.007)|-3.162| 35.54 |
>
> As shown in the table, for both SRIP and SO, the final value of $\lVert W^{\top}W-I\rVert_F$ is governed by the optimization process and the chosen scalar weight $w$. Unlike our $λ$-orthogonal regularization, these approaches do not provide direct control over $\lVert W^{\top}W-I\rVert_F$. Consistent with the reviewer's expectations, a weaker contribution of the regularizer to the total loss results in a diminished regularization effect on the backward transformation $B_λ$. When the scalar weight $w$ of the regularizer is reduced, the optimization process is unable to fully minimize the regularization term, particularly because competing loss components (such as MSE and the contrastive loss $L_C$) may favor a non-orthogonal transformation. For instance, when $w = 10^{-3}$ and $w = 10^{-2}$, the results obtained with SO, SRIP, and our $λ$-orthogonal regularization are comparable to those observed in the case of $λ = \infty$ (see Table 9), where the orthogonality constraint is entirely ignored. This occurs because, at such a small value of $w$, the contribution of the regularizer becomes negligible during optimization. To avoid this issue, in our method we set $w = 1$ for the λ-orthogonal regularization, thereby ensuring that the regularization term is effectively incorporated into the optimization process during the training of the backward transformation. This ensures that the regularization term achieves the target threshold λ, enabling precise control over the stability–plasticity trade-off in the backward transformation and leads to higher representation compatibility on the downstream task. As highlighted by the bold entries in the table, our method produces stable results (minor fluctuations are attributable to stochastic optimization) for $w = 1$ and $w = 10^{-1}$ in contrast to SO and SRIP. Conversely, when $w$ is very low ($10^{-2}$ or $10^{-3}$), the regularizer cannot be fully optimized, and our method behaves similarly to SO regularization, as our introduced constrains ($\lVert W^{\top}W-I \rVert_F\geqλ$) influences the minimum of the objective, which is never reached in practice. In contrast, due to its approximate formulation and greater complexity relative to SO, SRIP exhibits an even weaker regularization effect when $w$ is low.
>
> To further confirm the differences between SO and our approach, we increased the number of training epochs for the adapter from 200 to 600, employing a weight $w = 10^{-1}$ for both our method and SO. This extended training allows for more time to effectively minimize the regularization term. The results are shown in the following table.
>
> |Epochs|Method|$F(\phi_{\text{old}})/F(\phi_{\text{old}})$| $B_λ(\phi_{\text{new}})/F(\phi_{\text{old}})$| $B_λ(\phi_{\text{new}})/B_λ(\phi_{\text{new}})$ |ZS| Final Value of $\lVert W^{\top}W-I\rVert_F$|
> |-|-|-|-|-|-|-|
> |200|SO|59.11|69.56|74.88 (+3.106)|+0.022| 9.50 |
> |200|**Ours (λ=12)**|**59.93**|**70.70**|**75.20 (+3.419)**|**+0.076**| **12.12** |
> ||||||||
> |600|SO |58.44|69.01|73.38 (+1.606)|+0.034| 6.48 |
> |600|**Ours (λ=12)**|**60.01**|**71.13**|**75.12 (+3.339)**|**+0.096**| **12.02** |
>
> The table shows that SO, lacking an explicit constraint, keeps driving $\lVert W^{\top}W-I\rVert_F$ toward its minimum of 0, whereas our λ-orthogonal regularizer holds the value near the target threshold λ.

---

> > ### Comment · Reviewer_uQX7 · 2025-08-07
> >
> > Thanks to the author for providing the experiment. My concerns have been resolved.

---

> > > ### Author Response · Authors · 2025-08-07
> > >
> > > Thank you for your time in reviewing our manuscript. We are pleased that your concerns were addressed during the discussion period.

---

### Official Review · Reviewer_tsn4 · 2025-07-01

**Clarity:** 3
**Significance:** 3
**Originality:** 2
**Rating:** 5
**Confidence:** 3

**Summary:**

The paper proposes a new method to align the representation spaces of different models. The method leverages a smooth relaxation of the Heaviside function for a soft orthogonality constraint. The authors combine this regularization with 3 other losses: a forward loss, backward loss, and a contrastive loss for clustering. The paper shows that their method improves upon baselines for image retrieval and partial backfilling.

**Questions:**

1. In Table 1, what does Ind. Tr. mean? Is it expected that CMC and mAP of the last row of Ind. Tr. are the same as the last row of "Ours"?
2. Are there other alternatives to truncating the higher-dimensional features to match like a learned down projection (end of 3.2)?
3. Are there other applications of representation learning (besides retrieval) where soft orthogonalization could provide improvement?
4. In Appendix D, L_B seems to be the least important component of the loss and only results in a slight decrease, do you have a sense of why?
5. Line 921 in Appendix F should be deleted.

**Ethical Concerns:**

["NO or VERY MINOR ethics concerns only"]

**Final Justification:**

My final recommended score is a 5 (updated from a 4 before rebuttal) with confidence 3. The initial version of the paper had some problems with clarity of presentation as well as some concerns on method complexity and scope. During the rebuttal, the authors resolved the clarity issues, provided reasonable discussions on practical complexity and broader applicability, and conducted additional experiments on how their method performs under stronger model and distribution shift from ResNet to CLIP. These discussions resolved my main concerns, so I've updated my score.

**Limitations:**

Yes

**Quality:**

3

**Strengths And Weaknesses:**

Strengths:
1. The exposition and motivation for the method are clearly explained. There's a nice mix of toy settings and visualizations for showing the intuition of how affine and orthogonal transformations effect representation space.
2. The main method packages several ideas (soft orthogonal constraints, contrastive clustering, new backfilling) together into a new approach. The experiments are well-executed and support the paper's claims for improving retrieval compatibility and backfilling. The appendix also has a nice set of ablations for each component of the method.

Weaknesses
1. Some of the presentation of results are a bit unclear and hard to parse. Table 1 in particular would be easier to read if the best results were bolded / highlighted. Throughout section 4, the analysis of the experimental results would be more clear if specific numeric results were referenced (instead of just table numbers).
2. The general complexity of the method (and addition of a threshold) may make it harder to use in practice. The paper does reference these limitations and provide ablations on the main components.
3. The general scope of the paper seems quite specific to retrieval tasks; it's a bit unclear how many of the empirical results would be of interest or applicable for broader representation learning.

---

> ### Author Rebuttal · Authors · 2025-07-30
>
> We thank the Reviewer for their thoughtful feedback and for highlighting several key strengths of our work. We appreciate the recognition of our clear exposition and motivation, as well as the use of toy settings and visualizations to illustrate the intuition behind affine and orthogonal transformations. We are glad the integration of soft orthogonal constraints, contrastive clustering, and novel backfilling into a unified approach was noted, along with the strong experimental support and comprehensive ablations. Finally, we value the acknowledgment of our method’s relevance to model update compatibility and its seamless integration into existing adaptation pipelines.
>
> **[Suggestions for Clearer Presentation of Results in Table 1 and Section 4]**
>
> We thank the Reviewer for these suggestions to improve the clarity of our manuscript. In the revised version, we will highlight the best results in each table. Furthermore, in Section 4, we will reference specific results directly in the text rather than referring only to the table numbers, as recommended.
>
> **[Practical Complexity and Usability of the Proposed Method]**
>
> While our method may initially appear complex, it is, in fact, straightforward to implement in practice. The approach requires training only two matrices, resulting in a small number of parameters to optimize. Moreover, because our method operates solely on the extracted embeddings, it does not require any knowledge of the underlying models and is therefore applicable across different objectives, architectures, and types of learned representations (as demonstrated in the tables referenced in Reviewer SkaR's response).
>
> In contrast to previous methods, which either focus solely on alignment loss without any representation clustering loss (e.g., FCT), or require specific architectural components of the pretrained models (e.g., FastFill, which requires access to the classifier of the new model), our approach addresses these limitations. Additionally, while existing baselines provide only forward adaptation, our method is designed to achieve both forward and backward compatibility, thereby addressing practical needs that prior works do not meet. For instance:
>
> -  $B_{\perp}(\phi_{\text{new}})/F(\phi_{\text{old}})$ yields higher retrieval values (i.e., better compatibility) compared to the baselines.
> - $B_{\perp}(\phi_{\text{new}})/\phi_{\text{old}}$ can be achieved exclusively by our method. From a practical standpoint, this allows compatibility to be established even before all gallery items are forward-adapted using $F$.
> - Since our approach provides a unified representation space, even when the gallery is in a hybrid form (i.e., with some elements already adapted and others not), using $B_{\perp}(\phi_{\text{new}})$ still ensures compatibility. This is something that neither FCT nor FastFill can achieve.
>
>
> As discussed in the Limitations section (Appendix G), the hyperparameter $\lambda$ can be selected via cross-validation or a small hyperparameter search on a held-out subset of the downstream dataset. While automatic tuning of this parameter remains an open problem and a direction for future work—since we are the first to propose this form of relaxed regularization—the introduction of such a parameter is not uncommon. For instance, in CLIP training, the temperature parameter affects the training process and depends on the training dataset and architectures, yet it is not typically regarded as a limitation.
>
>
> **[Definition of “Ind. Tr.” in Table 1]**
>
> The term "Ind. Tr." stands for "Independently Trained." To avoid any potential confusion, we will change this label to "Ind. Train" in the revised manuscript. This row in each table represents the retrieval performance obtained by directly using the independently trained models, without any form of adaptation.
>
> **[Clarification on Identical CMC and mAP Values in “Ind. Tr.” and “Ours”]**
>
> This outcome is expected and intentional according to our methodology. This occurs specifically when a strict orthogonal transformation $B_{\perp}$ is applied to the new model representation. Since $B_{\perp}$ is an isometry, it preserves the geometry (angles and magnitude) of the new model’s embedding space and does not modify any relationships inside the representation. Therefore, the retrieval performance remains identical in both cases, as reflected in the last row of "Ind. Tr." and the last row of "Ours".
>
> **[Alternatives to Feature Truncation for Dimensionality Matching]**
>
> Yes, there are alternatives to truncating the higher-dimensional features to achieve dimensionality matching. One common approach is to apply zero padding to the smaller feature vector in order to match the dimension of the larger one. However, since the padded dimensions do not contain any information, especially in retrieval scenarios where cosine similarity is used, both truncation and padding yield equivalent results.
>
> Another possibility is to employ a learned down-projection using a rectangular matrix. However, to ensure a strict orthogonality (isometry), the transformation matrix must be square. Furthermore, as noted by [36], enforcing orthogonality on a rectangular matrix through soft orthogonality regularization can lead to suboptimal solutions.
>
>
> **[Broader Applicability Beyond Retrieval Tasks]**
>
> As it is demonstrated in [36], soft orthogonalization has been applied to regularize all the weights of a CNN during training, and could benefit from the increased plasticity offered by our proposed $\lambda$-orthogonal regularization. While retrieval is the standard scenario for evaluating compatibility [1], our approach is broadly applicable to any task that requires representation adaptation, as it focuses on model alignment and clustering of learned representations. As demonstrated in our downstream task adaptation experiments, our regularization approach yields improved performance compared to a strict orthogonal constraint, making it a valuable approach in domain adaptation scenarios as well. Furthermore, enforcing geometrical consistency while allowing adaptability has recently been investigated in the context of continual learning for multimodal training [2]. However, the authors of [2] promote this property indirectly through a knowledge consolidation loss, rather than by directly applying a regularization constraint. This highlights both possible future research and the potential applicability of our $\lambda$-orthogonal regularization across various areas of representation learning.
>
> [1] Shen, Y., Xiong, Y., Xia, W., & Soatto, S. (2020). Towards backward-compatible representation learning. In Proceedings of the IEEE/CVF Conference on Computer Vision and Pattern Recognition (pp. 6368-6377).
> [2] Liu, W., Zhu, F., Wei, L., & Tian, Q. (2024). C-CLIP: Multimodal continual learning for vision-language model. In The Thirteenth International Conference on Learning Representations.
>
> [36] Bansal, N., Chen, X., & Wang, Z. (2018). Can we gain more from orthogonality regularizations in training deep networks?. Advances in Neural Information Processing Systems, 31.
>
> **[Analysis of the Impact of $L_B$ in Appendix D]**
>
> To assess the specific impact of the $L_B$ loss, one should refer to the results reported in the $B(\phi_{\text{new}})/\phi_{\text{old}}$ column of Table 10 and 11. The $\mathcal{L}_B$ loss optimizes the alignment between the adapted new representation and the old representation via an isometric transformation $B$. Since the old representation is fixed (pre-extracted with the old model), it serves as a static reference to which the new model's representation is aligned through the optimization of $\mathcal{L}_B$.
>
> As the isometric transformation $B$ can only rotate or reflect the embedding space without altering its internal structure, the effect of $\mathcal{L}_B$ alone is limited compared to the other loss components, which promote intra-class clustering and employ transformations with higher flexibility, as discussed in Section 3.4.
>
> Furthermore, when $L_C$ is combined with either $L_B$ or $L_F$, it can be observed that the backward compatibility performance (i.e., $B(\phi_{\text{new}})/\phi_{\text{old}}$) is higher when utilizing $\mathcal{L}_B$ compared to $\mathcal{L}_F$. This is because alignment to the old representation is explicitly enforced by $\mathcal{L}_B$, whereas $\mathcal{L}_F$ does not directly target it. The combination with $\mathcal{L}_C$ further enhances intra-model alignment and backward compatibility performance.
>
>
> **[Removal of Line 921 in Appendix F]**
>
> We will remove the line in the revised version.

---

> > ### Comment · Reviewer_tsn4 · 2025-08-05
> >
> > Thank you for the detailed response! The rebuttal resolved most of my main concerns, so I updated my score accordingly. I encourage the authors to include some discussion on Method Complexity and Broader Applicability within the updated manuscript.

---

> > > ### Author Response · Authors · 2025-08-05
> > >
> > > Thank you for your time in reviewing our manuscript, as well as for your helpful feedback and updated score. We appreciate your suggestion and will include discussion on Method Complexity and Broader Applicability in the revised manuscript.

---

### Official Review · Reviewer_SkaR · 2025-07-02

**Clarity:** 3
**Significance:** 3
**Originality:** 3
**Rating:** 5
**Confidence:** 4

**Summary:**

The authors study the problem of embedding model compatibility in vector database search. If a vector database is generated with a model $\phi_{old}$ but later a model $\phi_{new}$ is trained, can we still make use of the existing embeddings without backfilling? The authors propose numerous strategies to ensure model compatibility. Their primary contribution is $\lambda$-orthogonality: during training of the new model, introduce a regularization to ensure that new representations are approximately an orthogonal transformation of the old model features. They also introduce strategies to ensure this does not hinder new model training. They show improvements in the forward and backward compatibility settings.

**Questions:**

Generally addressing weaknesses would improve my score. Here's an option:

Try applying these approaches to a CLIP model, do compatibility between old ImageNet1k and new CLIP model trained on a small dataset (CC12M). Even if you can't directly do this in the full backward compatibility scenario, I'm interested in how/if the modeling losses restrict new model performance under distribution/objective shift.

**Ethical Concerns:**

["NO or VERY MINOR ethics concerns only"]

**Final Justification:**

Authors addressed my weaknesses. In particular, the CC12M experiment showed improvement in larger domain updates.

**Limitations:**

yes

**Quality:**

3

**Strengths And Weaknesses:**

### Strengths

+ Interesting technical contributions:

1. $\lambda$-orthogonality loss (surprising that this does not damage new model loss too much)
2. Cluster based approach to enhance alignment as well (similar to BCT)

+ Training forward and backward compatibility at the same time yields a pretty flexible set of options for compatible vector search
+ Interesting study of partial backfilling along with forward transformations, which adds a good picture of efficient model updates.


### Weaknesses

- In comparison to other forward compatibility approaches, which are new model training agnostic, this adds substantially more losses to new model training, potentially limiting utility in practical training pipelines.
- Not clear how the intraclass clustering approach can be adapted to CLIP-like and other self-supervised learning approaches (e.g. Dino) models, which are very common in image vector databases these days.
- Not clear how the orthogonal projection would behave in more significant model update scenarios (e.g. a change in distribution/objective, with phi_old = Classifier on ImageNet1k and phi_new = CLIP on CC12M).

---

> ### Author Rebuttal · Authors · 2025-07-30
>
> We thank the Reviewers for their careful evaluation and for highlighting several key strengths of our work. In particular, we are glad that the orthogonality loss and the cluster-based alignment approach—both of which underpin our adaptation method—were found to be interesting. We also appreciate the acknowledgment of our efforts to enable both forward and backward compatibility during training, which yields a flexible set of options for compatible vector search. Additionally, we are pleased that the study of partial backfilling alongside the transformations was noted as providing a comprehensive perspective on efficient model updates. Below, we address each of the weaknesses raised in the review in detail.
>
>
> **[Practical Limitations Due to Additional Losses in New Model Training]**
>
> Thank you for raising this significant point. To clarify, we do not train the models themselves, but rather adapters; all backbone models remain frozen (see L103-104), consistent with both FCT and FastFill approaches (we utilize pretrained model weights, downloading them as independently trained models when available—see L295-296 and L884-887). These adapters are implemented as a single matrix, enabling fast training and ensuring practical applicability in real-world scenarios.
>
> We will add specific symbols to Figure 1 to distinguish the trainable adapters from the frozen model backbone, thereby avoiding any possible misunderstanding.
>
> Moreover, while FCT is the first work in this area, as discussed in our paper, it has certain limitations—specifically, its objective loss is solely the MSE between old and new representations. This enforces alignment of the old representation to the new one but does not effectively capture additional information from the adaptation training data. FastFill attempts to address this limitation by introducing an extra loss for improved forward transformation learning; however, this additional loss requires direct access to the new model, particularly its classifier, which is not always possible.
>
> In contrast, our method operates directly on the embeddings extracted from the networks, making our approach model-agnostic like FCT, but with the added advantage of being able to incorporate extra information from the adaptation data via a contrastive loss that enforces both intra-class clustering and inter-model alignment. Compared to FastFill, our method introduces only a single, simple loss (MSE) for training the backward transformation.
>
>
> **[Applicability of Intr-aclass Clustering and Orthogonal Projection in Modern Self-Supervised (DINOv2) and CLIP-like Models]**
>
> We thank the Reviewer for encouraging us to investigate update scenarios involving distribution or objective shifts, which will further validate our approach. Our method operates directly on embeddings extracted from any pretrained model; all backbone models remain frozen and are used solely to obtain embedding vectors. As a result, our approach is agnostic to the specific pretraining strategy (i.e., objective function) or dataset (i.e., data distribution).
>
> The orthogonality constraint imposed on the backward transformation ensures that the representations learned by the pretrained model are preserved during adaptation, thereby avoiding distribution shift, as this transformation is an isometry.
> In our experiments, we already consider the case where adapters are trained on embeddings generated from a downstream task dataset, whose data distribution typically differs from that of the dataset used for pretraining. Thus, our method can be readily applied to CLIP-like models and other self-supervised architectures such as DINOv2.
>
> However, challenges may arise if the new model’s representation quality is insufficient for a specific dataset. As discussed in our Limitation section (Appendix G), our method assumes that the new model’s embedding space is more expressive (e.g., exhibiting higher retrieval accuracy or stronger clustering) than that of the old model. If the updated model is not comparable or has lower representation quality—due, for example, to domain mismatch, limited training data, or architectural regressions—both the forward and backward adapters may fail to improve retrieval performance.
>
> To empirically investigate this scenario, as requested by the Reviewer, we conducted experiments using a ResNet-18 pretrained on ImageNet1K as the old model, and both a CLIP model pretrained on CC12M and a DINOv2 model (“vit_small_patch14_dinov2”) as the new models. The dataset used to train both the forward and backward transformations was ImageNet1K. This setup represents a considerable shift in both data distribution and model objective relative to the new models. Notably, FastFill cannot be applied in this context, as both CLIP and DINOv2 lack classifiers. For fairness, we used the same hyperparameters across all methods as in the main manuscript.
>
> In the following Table, we report the results obtained using DINOv2 as the new, independently trained model.
>
> | Method        | Query/Gallery                              | CMC-Top1 | mAP   |
> |---------------|--------------------------------------------|----------|-------|
> | **Ind. Tr.**  | $\phi_{\text{old}}/\phi_{\text{old}}$      | 55.62 | 26.91 |
> |               | $\phi_{\text{new}}/\phi_{\text{old}}$      | 0.04 | 0.17 |
> |               | $\phi_{\text{new}}/\phi_{\text{new}}$      | 71.92 | 44.07  |
> | **FTC**       | $F(\phi_{\text{old}})/\phi_{\text{old}}$   | 0.04 | 0.17 |
> |               | $F(\phi_{\text{old}})/F(\phi_{\text{old}})$| 59.33    | 37.53 |
> |               | $\phi_{\text{new}}/F(\phi_{\text{old}})$   | 67.97    | 41.07 |
> | **Ours**      | $F(\phi_{\text{old}})/\phi_{\text{old}}$   | 54.82    | 32.14 |
> |               | $F(\phi_{\text{old}})/F(\phi_{\text{old}})$| 61.30    | 41.95 |
> |               | $B_{\perp}(\phi_{\text{new}})/\phi_{\text{old}}$ | 58.73 | 31.50 |
> |               | $B_{\perp}(\phi_{\text{new}})/F(\phi_{\text{old}})$ | 68.74 | 43.78 |
> |               | $B_{\perp}(\phi_{\text{new}})/B_{\perp}(\phi_{\text{new}})$ | 71.92 | 44.07 |
>
>
> As shown in the Table, our method achieves compatibility even when DINOv2 is used as the new model. This demonstrates the applicability of our approach to models trained with different objectives and datasets.
>
> Moreover, as requested by the Reviewer, we present in the following Table the results obtained with FCT and our approach in the scenario where the old model is a ResNet-18 and the new model is a CLIP model pretrained on CC12M.
>
> | Method        | Query/Gallery                              | CMC-Top1 | mAP   |
> |---------------|--------------------------------------------|----------|-------|
> | **Ind. Tr.**  | $\phi_{\text{old}}/\phi_{\text{old}}$      | 55.62 | 26.91 |
> |               | $\phi_{\text{new}}/\phi_{\text{old}}$      | 0.04 | 0.17 |
> |               | $\phi_{\text{new}}/\phi_{\text{new}}$      | 44.29     | 16.15  |
> | **FTC**       | $F(\phi_{\text{old}})/\phi_{\text{old}}$   | 0.04 | 0.17 |
> |               | $F(\phi_{\text{old}})/F(\phi_{\text{old}})$| 42.58    | 16.93 |
> |               | $\phi_{\text{new}}/F(\phi_{\text{old}})$   | 42.96    | 16.88 |
> | **Ours**      | $F(\phi_{\text{old}})/\phi_{\text{old}}$   | 61.13    | 41.22 |
> |               | $F(\phi_{\text{old}})/F(\phi_{\text{old}})$| 57.69    | 41.08 |
> |               | $B_{\perp}(\phi_{\text{new}})/\phi_{\text{old}}$ | 30.02 | 16.68 |
> |               | $B_{\perp}(\phi_{\text{new}})/F(\phi_{\text{old}})$ | 44.93 | 29.26 |
> |               | $B_{\perp}(\phi_{\text{new}})/B_{\perp}(\phi_{\text{new}})$ | 44.29     | 16.15 |
>
> In this scenario, the pretrained CLIP model exhibits lower retrieval performance on ImageNet1K compared to ResNet-18. This is a well-known limitation of multi-modal training, where intra-modal misalignment can negatively impact the quality of single-modality representations [1]. Specifically, CLIP models are optimized for cross-modal retrieval rather than single-modality retrieval tasks, in contrast to DINOv2 or ResNet-18, which are trained exclusively on a single modality.
>
> [1] Mistretta, M., Baldrati, A., Agnolucci, L., Bertini, M., & Bagdanov, A. D. Cross the Gap: Exposing the Intra-modal Misalignment in CLIP via Modality Inversion. In The Thirteenth International Conference on Learning Representations.
>
> This reduction in performance of the new model relative to the old one causes FCT to degrade the overall retrieval capacity of the system, failing to achieve compatibility, as it attempts to transform the higher-quality representations of the old model into the lower-performing representations of the new model. In contrast, our method introduces an additional loss that encourages both intra-class clustering and inter-model alignment of feature representations on the specific training dataset. As a result, the transformation $F$, due to its greater flexibility, improves the performance of the old model’s representations. Even in this challenging scenario, our approach outperforms FCT, further validating the robustness of our method.
>
> The intra-class clustering loss defined in Eq. 8 relies on the availability of class labels to encourage embeddings from the same class to cluster together while pushing apart embeddings from different classes. In scenarios where class labels are not available, Eq. 8 naturally reduces to an unsupervised contrastive loss, similar to the objective used for training CLIP models. In this unsupervised setting, we contrast pairs of representations originating from different models, and clustering—since it cannot be enforced directly—becomes a byproduct resulting from embedding similarity. Consequently, our approach is flexible and can be applied in both supervised and unsupervised training scenarios, depending on the availability of labels for the downstream task.

---

> > ### Comment · Reviewer_SkaR · 2025-08-06
> > **Response**
> >
> > Thank you to the authors for the comprehensive response. I have no more complaints and I am happy to raise my score to a 5 (Accept). Please include these experiments in the final revision.
> >
> > Also a typo: FTC should be FCT in these tables and many tables throughout the main paper.

---

> > > ### Author Response · Authors · 2025-08-06
> > >
> > > Thank you very much for your comments, the suggested experiments, and the time taken to review our manuscript. We are glad that the additional experiments in our rebuttal were helpful. As suggested, we will incorporate them into the Appendix of the manuscript and correct the typo (FTC -> FCT) throughout the manuscript.

---

### Official Review · Reviewer_FJf7 · 2025-07-03

**Clarity:** 2
**Significance:** 3
**Originality:** 3
**Rating:** 4
**Confidence:** 2

**Summary:**

The paper focuses on the challenge of reconciling representations learned by different neural networks in retrieval systems, particularly given the high training costs and potential inconsistencies. It highlights two common approaches for adapting learned representations: affine transformations, which are adaptable but can distort the original representation, and orthogonal transformations, which preserve structure but are less adaptable. The core problem is how to align the latent spaces of updated models with previous ones on downstream distributions, all while preserving the newly learned representations. To address this, the authors propose a λ-orthogonality regularization applied during the learning of an affine transformation. This technique aims to achieve distribution-specific adaptation while simultaneously retaining the original learned representations. The paper validates this approach through extensive experiments across various architectures and datasets, demonstrating its ability to preserve zero-shot performance and ensure compatibility across model updates.

**Questions:**

Refer to the weakness.

**Ethical Concerns:**

["NO or VERY MINOR ethics concerns only"]

**Final Justification:**

I appreciate the rebuttal, and I think the updated manuscript should contain the revised notations that are provided here. That being said, given that I am not an expert in this area, I'd keep my positive rating.

**Limitations:**

I am not very familiar with this area, so unfortunately cannot point to any deeper limitations.

**Quality:**

3

**Strengths And Weaknesses:**

## Strengths

I think the idea of adapting representations is quite interesting as well as the problem is well motivated. Paper is more or less written in a easy fashion, although I've some questions which I point out to in the next section.

## Weakness
- Can authors add a quick proof of B being orthogonal if it is represented as a matrix exponential of a skew symmetric matrix?
- For the $\lambda$ orthogonality constraint, I am not sure on what parameters is it applied. While the equation refers to W, I am not sure where is this W coming from. From what I've understood, it could be the learnable parameters in B, but it is not clear.
- I am not following the contribution of individual loss terms. Given the best results mentioned in the experiments are when we just apply backward transformation on the new ones, for both query and gallery, can one not get away with just having backward compatibility loss?
- Table~10 last column seems off, as why should the performance remain same as 76.63 despite heavily changing the  objective functions.
- For Table~2 why should one see improvement over the independently trained new model when applied the backward compatibility mapping?

---

> ### Author Rebuttal · Authors · 2025-07-30
>
> We sincerely appreciate the Reviewers’ thoughtful feedback and their recognition of our contributions to representation compatibility through our adaptation approach and its underlying motivation. Below, we address each of the points raised in the review in detail.
>
> **[Proof of Orthogonality for Exponentials of Skew-Symmetric Matrices]**
>
> We thank the Reviewer for this suggestion. To clarify why $B$ is orthogonal when reparametrized as the exponential of a skew-symmetric matrix, we provide the following proof.
>
> > **Proof.** Let $P$ be any real skew-symmetric matrix ($P^T = -P$) and define $B$ as its exponential:
> >
> > $$
> > B = e^P = \sum_{k=0}^\infty \frac{P^k}{k!}\,.
> > $$
> >
> > Since transposition and the matrix exponential commute, it follows that
> >
> > $$
> > B^T = \bigl(e^P\bigr)^T = e^{P^T} = e^{-P}\,.
> > $$
> >
> > Therefore
> >
> > $$
> > B^T B \;=\; e^{-P}\,e^P \;=\; e^{-P + P} \;=\; e^0 = I\,,
> > $$
> >
> > showing that $B^T B = I$ and hence $B$ is orthogonal.
> > $\Box$
>
> This demonstrates that any matrix of the form $e^{P}$ is orthogonal.
>
>
> **[Clarification of $W$ and the Learnable Parameters of Transformation $B$]**
>
> The Reviewer is correct. $W$ denotes the weight matrix of the transformation $B_\lambda$ (see Lines 148–149): $B_\lambda: \mathbb{R}^n \to \mathbb{R}^n,\quad B_\lambda(x) = W x + b$, where $W \in \mathbb{R}^{n \times n}$ and $b \in \mathbb{R}^n$.
>
> For clarity regarding the learnable parameters of the backward transformation, we provide additional details for Sections 3.2 and 3.3 below.
>
> In Section 3.2, we enforce strict orthogonality on the transformation $B_\perp$ by reparameterizing $W$ as the matrix exponential of a skew-symmetric matrix $P$; thus, the learnable parameters are the upper triangular elements of $P$.
>
> In contrast, in Section 3.3, the transformation $B_\lambda$ uses a general weight matrix $W \in \mathbb{R}^{n \times n}$, and we encourage orthogonality via a a $\lambda$-orthogonality regularization:
>
> $$
> \min_{W} \|W^T W - I\|_F \quad \text{s.t.} \quad \|W^T W - I\|_F \geq \lambda
> $$
>
> If the Reviewer considers it necessary, we are willing to revise the manuscript to avoid any potential confusion by changing the symbol $W$ and referring directly to $W_B$, as it is the transformation subject to the $\lambda$-orthogonality regularization. Below, we provide an example reflecting this change:
>
> $$
> \min_{W_B} \|W_B^T W_B - I\|_F \quad \text{s.t.} \quad \|W_B^T W_B - I\|_F \geq \lambda
> $$
>
> **[Contribution of Individual Loss Terms and Practical Implications of Learned Transformations]**
>
> We thank the Reviewer for this insightful question. While it may initially appear that training only the backward adaptation loss (Eq. 2) is sufficient—since it directly encourages alignment of new queries with the old gallery—in practice, this approach is limited. The contribution of each loss term is analyzed in detail in Appendix E of the manuscript. As shown in Tables 10 and 11, optimizing only the backward loss $L_B$ results in a significant drop in performance for the $B(\phi_{\text{new}})/F(\phi_{\text{old}})$ scenario. This degradation arises because the forward transformation $F$ remains untrained when only $\mathcal{L}_B$ (see Eq. 2) is optimized, which in turn prevents effective retrieval in settings where the transformation $F$ is involved.
>
> As noted by the Reviewer, the best results are always obtained when both the gallery and query sets are encoded with the newly trained independent model ($\phi_{\text{new}}/\phi_{\text{new}}$ or $B_{\perp}(\phi_{\text{new}})/B_{\perp}(\phi_{\text{new}})$). However, this scenario is rarely practical, as extracting embedding vectors for the gallery set is the most computationally expensive operation. This process corresponds to the re-indexing discussed in previous works, which is typically avoided by leveraging representation compatibility. Thus, the case where both gallery and query are re-indexed with the new model serves as an upper bound for achievable system performance following complete re-indexing.
>
> In practical systems, the query set is usually much smaller than the gallery. Applying the backward transformation $B$ only to the query set enables compatibility with the existing gallery and improves overall system performance, while avoiding the costly re-indexing of the gallery ($B_{\perp}(\phi_{\text{new}})/\phi_{\text{old}}$ and $B_{\perp}(\phi_{\text{new}})/F(\phi_{\text{old}})$). Furthermore, applying the forward transformation $F$ to the already extracted old gallery embeddings introduces minimal computational overhead [25], making it feasible for real-world deployments. This approach also yields better retrieval performance compared to using the untransformed old gallery embeddings directly. Given these considerations, the most practical scenario is $B(\phi_{\text{new}})/F(\phi_{\text{old}})$, which should be directly compared with the $\phi_{\text{new}}/F(\phi_{\text{old}})$ results reported by the other baselines.
>
> To better illustrate the relevance of each transformation, we highlight the most informative cases:
>
> - $F(\phi_{\text{old}})/\phi_{\text{old}}$: Represents the retrieval performance when only the query set is forward-transformed and compared against the old gallery.
>
> - $F(\phi_{\text{old}})/F(\phi_{\text{old}})$: Indicates performance when both the gallery and query are forward-transformed using $F$.
>
> - $B(\phi_{\text{new}})/\phi_{\text{old}}$: Shows performance when only the query set is backward-transformed. (It is worth noting that only our approach allows for this scenario.)
>
> - $B(\phi_{\text{new}})/F(\phi_{\text{old}})$: Reflects the performance when the query set is backward-transformed and the old gallery is forward-transformed.
>
> Finally, in the context of downstream task adaptation, even $B(\phi_{\text{new}})/B(\phi_{\text{new}})$ provides valuable insight. Specifically, it demonstrates an improvement over $\phi_{\text{new}}/\phi_{\text{new}}$, indicating that relaxing the strict orthogonality constraint enables slight adaptability to the downstream task of the pretrained representation of the new independently trained model while preserving its zero-shot performance on the ImageNet1K dataset.
>
> **[Unchanged Performance in Table 10 Despite Objective Changes]**
>
> We thank the Reviewer for raising this question. As the Reviewer correctly observed, the retrieval performance in Table 10 for the $B_{\perp}(\phi_{\text{new}})/B_{\perp}(\phi_{\text{new}})$ case remains unchanged (76.63), despite significant changes in the objective functions.
>
> This result is expected and consistent with the backward transformation proposed in our approach. Since $B_{\perp}(\cdot)$ is a strict orthogonal transformation—enabled by the exponential reparameterization, as detailed in Section 3.2—this transformation is an isometry, meaning it preserves the geometric structure (angles and magnitudes) of the new model representation. Consequently, the performance of the new independent model is maintained under this transformation regardless of the optimized objective function. This behaviour is also present in Tables 1, 6, and 7, where $B_{\perp}(\phi_{\text{new}})/B_{\perp}(\phi_{\text{new}})$ is comparable to $\phi_{\text{new}}/\phi_{\text{new}}$. The strict orthogonality is indeed a crucial property of the backward transformation $B$ when the geometric structure of a representation needs to be maintained.
>
> Only in the setting of downstream tasks, we relax this strict orthogonality with our $\lambda$-orthogonal regularization, allowing for slight adaptation of the new model representation to the specific task without compromising the original learned representation distribution, as explained in Section 3.3.
>
> **[Explanation for Improved Performance With Backward Compatibility Mapping in Table 2]**
>
> Thank you for your question. The improvement observed in Table 2 is attributable to the $\lambda$-orthogonal regularization applied to the backward compatibility transformation $B_\lambda$. This regularization enables the representation extracted by the new model to effectively adapt to new data, thereby improving its downstream performance.
>
> This can be attributed to the following reasons:
> - First, our $\lambda$-orthogonal regularization provides a granular control on the degree of alignment between the new and old model representations. It strikes a balance between strict orthogonality, which may overly constrain the new model representation, and no regularization, which may permit harmful drift in the representation. This controlled flexibility enables the backward transformation to preserve the original learned representation while also facilitating effective adaptation to new data.
> - Second, the backward compatibility mapping, which is also jointly optimized with the contrastive loss described in Eq. 8, encourages both intra-class clustering and inter-model alignment on the downstream task. This leads to better retrieval performance. As shown in the ablation study in Table 11, when the contrastive loss is not included in the optimization, the performance of $B_{\lambda}(\phi_{\text{new}})/B_{\lambda}(\phi_{\text{new}})$ is not always higher than that of $\phi_{\text{new}}/\phi_{\text{new}}$ (which is 71.78).
>
> Empirically, as shown in Table 2 and discussed in Appendix D, applying our $\lambda$-orthogonal regularization leads to superior adaptation to downstream tasks, compared to both strict orthogonality and the absence of regularization. Table 9 further illustrates the different behaviors of the backward transformation under different values of $\lambda$. Figure 5 demonstrates that allowing excessive deviation from orthogonality can degrade the learned distribution of the newly pretrained model (e.g., the increment of zero-shot performance on ImageNet drops below zero), while moderate regularization improves downstream retrieval performance.

---

> > ### Comment · Reviewer_FJf7 · 2025-08-04
> > **Thanks for the rebuttal!**
> >
> > I appreciate the rebuttal, and I think the updated manuscript should contain the revised notations that are provided here. That being said, given that I am not an expert in this area, I'd keep my positive rating.

---

> > > ### Author Response · Authors · 2025-08-05
> > >
> > > Thank you very much for your feedback and for taking the time to review our manuscript. We are glad that the clarifications and revised notations in our rebuttal were helpful. As suggested, we will incorporate them into the updated manuscript to further improve its clarity.

---

### Note · Authors · 2025-08-13

We sincerely thank the Reviewers and the AC for their insightful feedback and constructive discussions.

We appreciate the Reviewers’ recognition of our study’s key strengths: its clear motivation and exposition; the technical novelty of the proposed λ-orthogonality regularization, cluster-based alignment, and partial-backfilling strategy—each supported by detailed ablations; and rigorous empirical validations. Reviewers further commended the method’s conceptual simplicity, ease of integration, and practical relevance for retrieval systems requiring seamless model updates.

During the discussion phase, we addressed all raised concerns through detailed clarifications:

1. We present an analytical proof showing that any matrix expressed as the exponential of a skew-symmetric matrix is orthogonal, and refine the notation to eliminate ambiguity in the parametrization of the backward transform. We also clarify the distinct roles of the forward, backward, and contrastive loss terms—highlighting their respective effects on each transformation when used individually or in combination.

2. We provided additional experimental results for update scenarios involving data distribution or objective shifts, demonstrating that our approach is agnostic to the specific pre-training strategy or dataset. In particular, we conducted experiments using a CLIP and a DINOv2 model as the new models, obtaining superior performance compared to existing methods.

3. We clarified the broader applicability of our method beyond retrieval tasks, along with its practical complexity and usability. We train lightweight matrix adapters rather than the backbone (kept frozen), enabling rapid deployment.

4. We further contextualized our work within the literature, emphasizing that the novelty of the λ-orthogonality regularization lies in enabling controlled flexibility when adapting a model to downstream tasks, in contrast to methods such as Spectral Restricted Isometry Property (SRIP) and Soft Orthogonality (SO), which lack such control. We clarified our relationship to Canonical Correlation Analysis and Procrustes analysis: although inspired by the latter, our method permits slight modifications to representation geometry via the backward transformation under λ-orthogonality regularization—essential for improving performance on downstream tasks.

These clarifications have further strengthened our manuscript, providing a valuable and practical solution to model updates in retrieval systems.

---

### Decision · Program_Chairs · 2025-09-17

**Decision:**

Accept (poster)

**Comment:**

This paper tackles the important and practical problem of ensuring compatibility across independently trained neural network representations, with a focus on retrieval systems that require seamless model updates. The main contribution is a $\lambda$-orthogonality, which relaxes strict orthogonality in transformation learning to balance stability and adaptability. The authors propose joint learning of forward and backward transformations, coupled with contrastive clustering, to improve representation alignment. Experiments across diverse architectures and datasets -- including challenging distribution shifts (ResNet $\rightarrow$ CLIP, DINOv2) -- demonstrate the effectiveness of the method.

This paper makes a solid and practically relevant contribution to representation compatibility in retrieval systems. While the relation to prior work and the scope of applicability could be further elaborated in the final revision, the authors have adequately addressed these points during the rebuttal.